# The neural computation of inconsistent choice behavior

Vered Kurtz-David[1], Dotan Persitz[1], Ryan Webb[2,3] & Dino J. Levy [1,4]

Humans are often inconsistent (irrational) when choosing among simple bundles of goods, even without any particular changes to framing or context. However, the neural computations that give rise to such inconsistencies are still unknown. Similar to sensory perception and motor output, we propose that a substantial component of inconsistent behavior is due to variability in the neural computation of value. Here, we develop a novel index that measures the severity of inconsistency of each choice, enabling us to directly trace its neural correlates. We find that the BOLD signal in the vmPFC, ACC, and PCC is correlated with the severity of inconsistency on each trial and with the subjective value of the chosen alternative. This suggests that deviations from rational choice arise in the regions responsible for value computation. We offer a computational model of how variability in value computation is a source of inconsistent choices.

---

[1] Coller School of Management, Tel Aviv University, 55 Haim Levanon street, Ramat Aviv, Tel Aviv-Yafo, Israel 6997801. [2] Rotman School of Management, University of Toronto, 105 St George St, Toronto, ON M5S 3E6, Canada. [3] Department of Economics, University of Toronto, 50 St George St, Toronto, ON M5S 3G7, Canada. [4] Sagol School of Neuroscience, Tel Aviv University, 55 Haim Levanon street, Ramat Aviv, Tel Aviv-Yafo, Israel 6997801. Correspondence and requests for materials should be addressed to D.J.L. (email: dinolevy@post.tau.ac.il)

A fundamental axiom in neoclassical theories of choice behavior is that the decision-maker is consistent in her choices. For example, if a decision-maker chooses some combination of milk and cookies (bundle A) over another combination of milk and cookies (bundle B) and also chooses bundle B when a third bundle C was available, then—if she is consistent in her choices—she should not strictly prefer bundle C over bundle A in any subsequent choice. Consistency is the fundamental axiom underlying rational behavior and the neoclassical construct of utility maximization[1].

The study of consistency was formalized with the Generalized Axiom of Revealed Preference (GARP)[2]. Despite the centrality of rational behavior in neoclassical economics, since at least the 1950s studies have demonstrated that the choice behavior of humans violate consistency when choice sets are manipulated or framed[3–6]. Such results have given rise to a behavioral approach to decision-making, in which agents are not strictly treated as consistent or rational[4]. Explanations for such anomalies establish that the human decision process is limited by, or maladapted to, the particular choice context under the study. For example, agents might simplify the choice process by using various heuristics[6–8], misunderstand the problem[9], or in some cases, inconsistency might arise due to a limited cognitive capacity[10].

However, more recently, lab experiments and real consumption data suggest that a degree of choice inconsistency might be present in human decision-making, even absent any particular framing or context induced by the experimenter. Subjects are often inconsistent and violate GARP[11–15] even when choosing over simple bundles; for example, simply switching their choices when presented with the same lotteries[16,17]. Therefore, a degree of inconsistent behavior may be fundamental to the human decision-making process. Indeed, economic theorists have proposed that the valuations which underlie choice may themselves be stochastic[18,19]. By design, these theories place weak constraints on the pattern of choice behavior because utilities are typically assumed to be unobservable. Neuroscientific methods, on the other hand, suggest a stronger test of this hypothesis[20]. Many previous studies have identified several brain regions—primarily the ventral striatum (vStr), the ventromedial prefrontal cortex (vmPFC) and the posterior cingulate cortex (PCC)—that correlate with a utility function fit to choice behavior irrespective of the reward type[21–23]. Whether the value representations in these areas obey consistency, and how this network might give rise to inconsistent choice behavior, is unknown.

Similar to sensory perception and motor output, we propose that a substantial component of inconsistent behavior is due to variability in the neural computation of value. There is ample evidence demonstrating that both the behavioral and neural responses to the same sensory input are variable[24–26]. The predominant explanation for this phenomenon is due to a fundamental property of the nervous system—the inherent variability in neural activity[27,28]. We propose that the same variability is responsible for inconsistency in choice behavior when the framing or the choice context is stable. Our modeling approach is an application of the Random Utility framework[18,20], a parsimonious account of the aggregation of value signals and neural variability in the course of a decision[29–32]. In our model, valuations of choice options are inherently stochastic, and the skewed nature of neuronal activity[33] implies that the severity of an inconsistent choice results from larger fluctuations in the computation of value. This prediction might seem surprising. After all, one might not expect that a brain region which computes the valuations of choice alternatives is more active when those valuations are contradicted in an inconsistent choice. To asses this hypothesis, we measure the severity of an inconsistent choice on a trial-by-trial basis, and identify the neural correlates of inconsistent choices.

Measuring the severity of inconsistent choice presents a challenge for the analysis of the neural data. Existing methods for measuring inconsistency either count the number of GARP violations or estimate the extent of (hypothetical) changes to the dataset required to make choices consistent[14,15,34–36]; therefore, they all assign one inconsistency score per subject. This aggregation is severely limited because it yields only a simple between-subject analysis based on the average neural activity over all choices. By construction, it ignores the trial-by-trial variation in both behavior and neural activity. Such variation provides information not just about which subjects are more inconsistent, but also the level of activity in different brain areas during an inconsistent choice. To overcome this limitation, we develop a trial-specific inconsistency index, which measures the severity of inconsistency contributed by each choice. We apply our novel index to a well-established choice task, and use it to explore the neural computation underlying inconsistent choices on a trial-by-trial basis.

A few neuroscientific studies have previously examined consistency, albeit using only aggregate-level inconsistency indices. Patients with lesions in the medial prefrontal cortex (mPFC) violate GARP and transitivity more often than non-lesioned controls, suggesting this region is necessary for consistent behavior[37,38]. GARP violations increase with aging and are negatively correlated with gray-matter volumes in the ventrolateral prefrontal cortex[39]. Neural correlates of intransitive lottery choices have been observed in BOLD signal from the vStr, anterior cingulate cortex (ACC) and the dorsolateral prefrontal cortex (dlPFC)[40]. Although these studies have identified which brain regions are involved in inconsistent choices across subjects, no study has yet examined how such behavior might arise in healthy human brains on a trial-by-trial basis.

Therefore, we examine, using fMRI, the neural basis of choice inconsistency on each trial of a choice task. Importantly, in our task, subjects make choices over lotteries holding the framing fixed. Using our novel trial-specific inconsistency index, this design allows us to assess the degree of a violation of inconsistency on each trial. We then search for correlates to this index in the BOLD signal, with both a whole-brain analysis and a region of interest (ROI) analysis of brain regions known to participate in value-based choice: the vmPFC[21–23], the vStr[21,22], the PCC[22,41], and the ACC[42], which is also related to choice difficulty, foraging, control, and monitoring[43,44]. We find that the BOLD signal from these regions correlates with both inconsistency levels and utility. These findings are consistent with our computational model explaining how inconsistent choice behavior might arise in value-related regions.

## Results

**A novel trial-specific inconsistency index.** For a systematic search for the neural computations that give rise to inconsistent choices, we propose an index to measure the severity of inconsistency on a given trial. Our novel index is based on a Leave-One-Out procedure (Fig. 1b) applied to the Money Metric Index (MMI), first introduced by Halevy et al.[45]. The MMI is a parametric measure of the extent of GARP violations in a dataset of choices from linear budget sets. It measures the minimal adjustments (in percentages) of the budget lines required to reconcile the decision-maker's choices with the best-fitting parametric utility function (see the Methods section and Fig. 1a).

For each trial, our index (trial-specific MMI) calculates the difference between the Aggregate MMI index (calculated over all observations) and the MMI index calculated over all observations less the given trial. Hence, our trial-specific MMI index measures the severity of inconsistency per trial within a subject

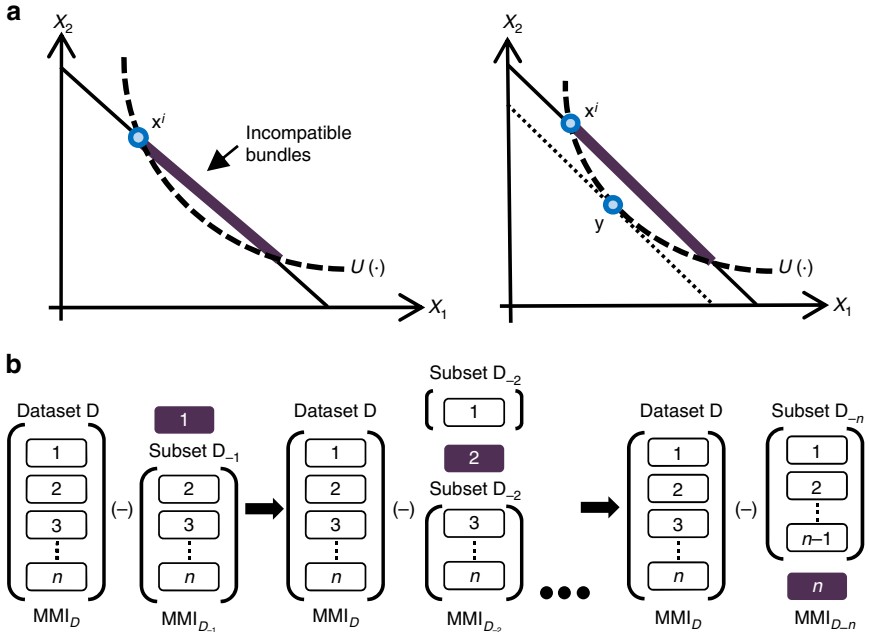

**Fig. 1** A trial-specific inconsistency index. **a** Computation of the MMI. Let $u(\bullet)$ be some utility function and let $\mathbf{x}^i$ be the bundle chosen by the subject in the trial $i$. A utility function induces a complete ranking on the bundles, i.e., if $u(\mathbf{x}) > u(\mathbf{y})$, then the bundle $\mathbf{x}$ is ranked above bundle $\mathbf{y}$. Also, by the revealed preference principle[72], when the subject chooses bundle $\mathbf{x}^i$, she reveals that she ranks this bundle over all other feasible bundles. These two rankings may be incompatible as shown in the figure: $u(\bullet)$ ranks all the bundles in the purple bold interval as better than bundle $\mathbf{x}^i$, while the subject ranked $\mathbf{x}^i$ as her most desired bundle in the given budget set. The extent of this incompatibility is measured by computing the maximal expenditure for which the two rankings agree (the minimal parallel inward adjustment of the budget line). Given the utility function $u(\bullet)$, we use average sum of squares to aggregate these adjustments over all observations. For a set of utility functions $\mathcal{U}$, we choose the utility function $u(\bullet)$ for which the aggregate adjustment is minimal. We refer to the aggregate adjustment of this $u(\bullet)$ as the MMI for the entire dataset $\mathbf{D}$ given the set of utility functions $\mathcal{U}$. **b** Leave-One-Out procedure. Denote the dataset by $\mathbf{D}$. Denote the dataset that is generated by removing observation $i$ from dataset $\mathbf{D}$ by $\mathbf{D}_{-i}$. For each observation $i$, the index is the difference between the aggregate index $\mathrm{MMI}_D$ calculated for the entire dataset $\mathbf{D}$ and the aggregate index $\mathrm{MMI}_{D_{-i}}$ calculated for the partial dataset $\mathbf{D}_{-i}$. Formally, the index for observation $i$ is $\mathrm{MMI}_D - \mathrm{MMI}_{D_{-i}}$

(see the Methods section). We then use it as a regressor to track the neural correlates of choice inconsistency.

A key benefit of the Aggregate MMI index is that it also yields parameter estimates of the subject's utility function. These subject-specific parameters may be used to estimate the subjective value (SV) assigned to the chosen bundle in each trial. We then use these SVs as another parametric regressor to identify the neural correlates of value modulation. Since SV does not depend on the specifics of the budget line (relative prices and endowment), while the trial-specific MMI depends on both, there is orthogonal information in the two regressors. Simultaneous identification of the severity of inconsistency on each trial, and the SV of that same trial, enables us to probe for the neural correlates of inconsistent choice behavior.

**Behavior**. Subjects made choices from linear budgets in the context of risk (following Choi et al.)[13] inside the fMRI scanner. On each trial, subjects were presented with a set of 50/50 lotteries between two accounts, $X$ and $Y$, and were asked to choose their preferred lottery (bundle). All possible lotteries in a given trial were represented along a budget line. The price ratios (slopes of the budget lines) and endowments were randomized across trials and subjects (Fig. 2, see the Methods section).

In line with previous literature, none of the subjects' choices satisfied GARP. However, there was considerable evidence that the subjects understood the task and did not behave randomly (Fig. 3a, c, Fig. 4a and Supplementary Note 2). Some predominant patterns in behavior are depicted in Fig. 4a (see Supplementary Figure 7 for scatterplots of all subjects). Figure 3b presents the recovered utility parameters in the sample (see Supplementary Table 6 for the

individual recovered parameters). A comparison with the Choi et al.[13] study reveals that the distributions of the Afriat inconsistency index (Methods and Supplementary Note 1)[2,34] are quite similar (Fig. 3c).

The parametric Aggregate MMI index is highly correlated with existing aggregate nonparametric indices (Fig. 3d, Afriat index: Spearman's $\rho = 0.538$, $p < 0.001$; Supplementary Figure 3a, the number of GARP violations: $\rho = 0.703$, $p < 0.001$), suggesting that, although parametric, the Aggregate MMI is a good measure of inconsistency[45]. Compared with nonparametric indices, there was considerable variability in the trial-specific MMI, therefore it can be used as a trial-by-trial regressor for neural activity (Fig. 4b).

**Neuroimaging**. We identify brain areas that correlate with the severity of inconsistency. A random-effect generalized linear model on the BOLD signal revealed that trial-specific MMI was positively correlated with activations in the mPFC and ACC ($p < 0.0005$, cluster-size corrected, Fig. 5a). This suggests that higher activation in these brain areas is correlated with more severe inconsistent choices on a given trial.

To verify that these areas also track value in our task (as previously reported[21–23]), we calculated the SV of the chosen bundle on each trial for each subject. The SV regressor was also positively correlated with activations in the mPFC and ACC ($p < 0.0005$, cluster-size corrected, Fig. 5b). A conjunction analysis revealed that both inconsistency and value modulations were correlated with activation in the mPFC and ACC (Fig. 5c, d, 1456 overlapping voxels, 28.7% of the SV cluster). This substantial overlap suggests that the neural computations that give rise to

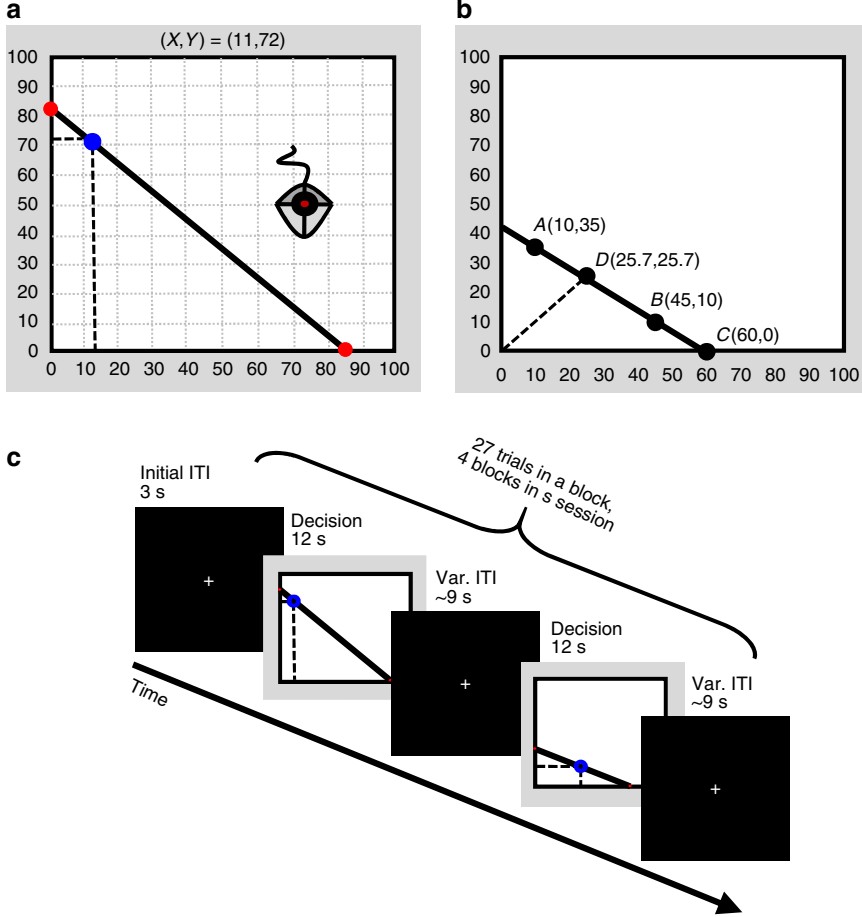

**Fig. 2** Experimental task (following Choi et al.[13]). **a** A Trial: Subjects were presented a visualization of a budget line with 50/50 lotteries between two accounts, labeled X and Y. Each point on the budget line represents a different lottery between the X and Y accounts. Subjects used a trackball to choose their preferred lottery (a bundle of the X and Y accounts) out of all possible lotteries along the line. For example, as depicted in Fig. 2a, the bundle (11,72) corresponds to a lottery with a 50% chance to win 11 tokens (account X), and a 50% chance to win 72 tokens (account Y), where 1 token equals 5 NIS (~$1.5). We varied and randomized the budget lines (slopes and endowments) across trials and subjects. At the end of the experiment, one trial was randomly selected as well as one of the accounts. The subject received the tokens she had allocated to the selected account in the selected trial. **b** Behavioral example: Given this budget line, the subject could choose A (10,35) with a 50% chance of winning 10 tokens (account X) and 50% chance of winning 35 tokens (account Y); or similarly, B (45,10). An extremely risk-seeking subject would choose C, the lottery with the maximal expected payoff, yielding a 50% chance of winning 60 tokens and a 50% chance of winning nothing. By contrast, an extremely risk-averse subject would choose D, the intersection with the 45-degree line. This bundle is a degenerate lottery, which allocates the same number of tokens for both X and Y accounts. **c** Timeline: Inside the scanner, subjects had a maximum of 12 s time window to make their choice, followed by a 9 s variable ITI. If subjects made a choice before the end of the 12 s time window, the remaining time was added to the ITI. There were 27 trials in each block, 4 blocks, for a total of 108 trials. Subjects completed a pre-scan questionnaire (see Supplementary Note 7 for an English version) and a practice block with a trackball outside the scanner to make sure the instructions and procedures were clear

inconsistent choices, hence deviations from rational choice, are related to the neural computations of value.

To increase the power of our analysis, we repeated our analysis on specific ROIs. Based on existing literature, we examined the vmPFC[21–23], vStr[21,22], dACC[42], and PCC[22,41], to test if choice inconsistency is related to value-based circuits. We also examined V1 as a control area. The ACC, vmPFC, and PCC were positively correlated with trial-specific MMI ($p$(Bonferroni) $< 0.05$, in the vmPFC and ACC, $p < 0.0005$, cluster-size corrected in PCC) and SV ($p$(Bonferroni) $< 0.05$, in the vmPFC and ACC, $q$(FDR) $< 0.05$ in the PCC), though we did not find any significant activation in the vStr. As expected, V1 was not correlated with trial-specific MMI (Fig. 5e), suggesting that only value-related regions are involved with choice inconsistency. These results corroborate the whole-brain analysis.

To verify that this group-level overlap reflects an overlap at the single-subject level, we also searched for conjunct areas on a subject-by-subject basis. In 24 of 33 subjects, there was a conjunct region between trial-specific MMI and SV in one or more of the ROIs: vmPFC, dACC, and PCC (Fig. 5f, Supplementary Table 5). The hypothesis that the subject-specific mean effects ($\beta$ values) for SV and trial-specific MMI were the same in the vmPFC and dACC could not be rejected (Wilcoxon sign-rank test, $p = 0.0504$ and $p = 0.2877$, respectively, multiple comparison corrected). However, for the PCC ROI, the difference was significant ($p < 0.005$). This suggests that the SV and trial-specific MMI predictors both have an important effect on the BOLD signal in the vmPFC and dACC (see Fig. 5g).

**Motivation for using trial-specific MMI.** To demonstrate the power and necessity of our trial-specific analysis, we also assessed whether a standard between-subject analysis using aggregate indices could identify the same brain areas found in our trial-by-trial analysis. We did not find significant correlations between any

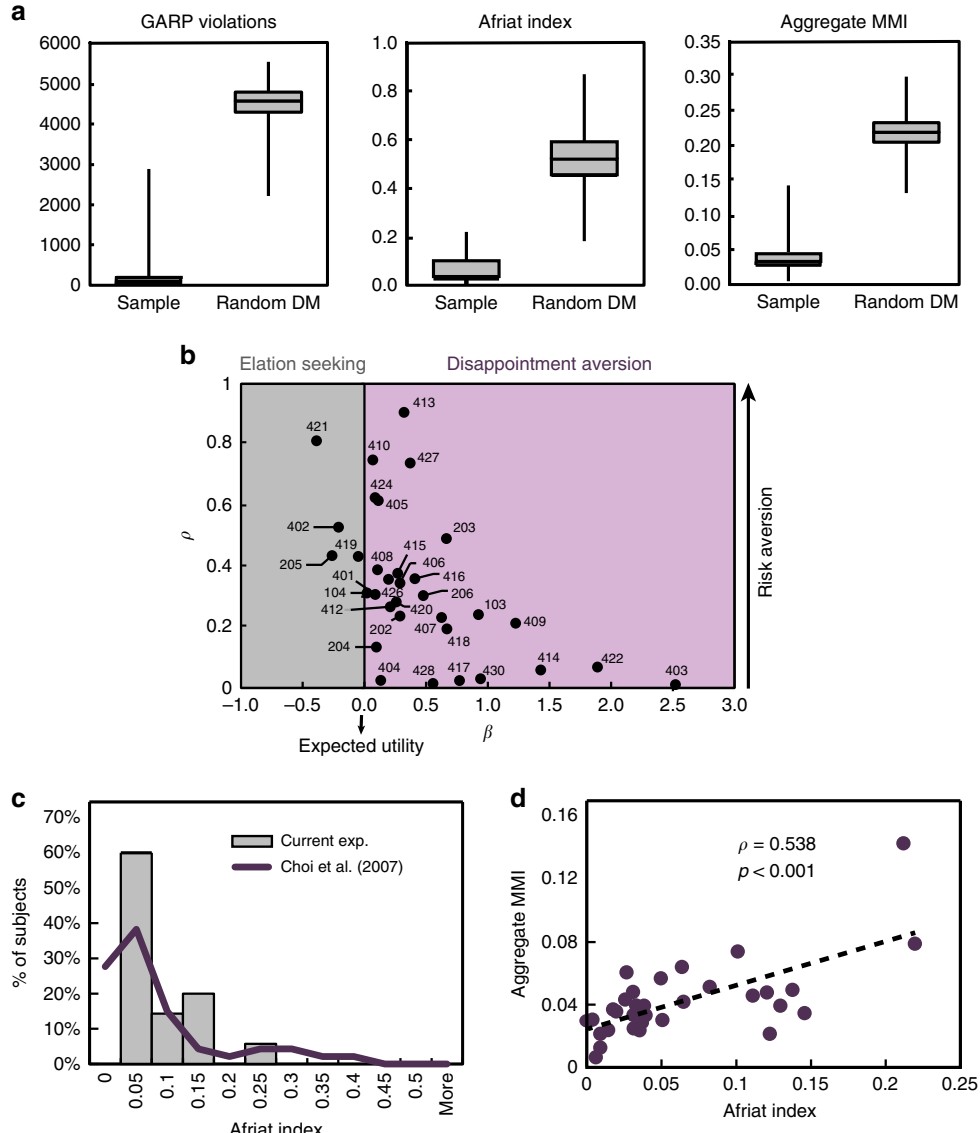

**Fig. 3** Behavioral results. **a** Actual subjects vs. random subjects: In our sample, the median number of GARP violations was 59 (std = 506.24, min = 1, lower quartile = 11, upper quartile = 166, max = 2873), the median Afriat index was 0.0362 (std = 0.058, min = 0.0007, lower quartile = 0.0258, upper quartile = 0.1011, max = 0.2197), and median Aggregate MMI score was 0.037 (std = 0.0242, min = 0.0065, lower quartile = 0.0297, upper quartile = 0.0481, max = 0.143). Both the aggregate MMI and Afriat index range between 0 (fully consistent) and 1. For comparison, 25,000 simulations of random decision-makers yielded median number of GARP violations of 4,568.5 (std 397.6, min = 2216, lower quartile = 4294, upper quartile = 4794, max = 5565), a median Afriat Index of 0.5229 (min = 0.182, lower quartile = 0.4545 upper quartile = 0.5905, max = 0.872) and a median Aggregate MMI of 0.2186 (min = 0.1332, lower quartile = 0.2047, upper quartile = 0.2326, max = 0.299). These high scores indicate that our subjects did not choose randomly along the budget lines. **b** Recovered parameters using the MMI with the Disappointment Aversion (DA) utility function with Constant Relative Risk Aversion (CRRA) functional form[69] (see Methods). The $\beta$ and $\rho$ parameter estimates for each subject are plotted. When $\beta < 0$ ("elation seeking", four subjects, gray area), the subject overweights the higher prize. When $\beta < 0$ ("disappointment aversion", 29 subjects, purple area), the subject overweights the lower prize. Higher $\rho$ values represent higher risk aversion levels. When $\beta < 0$, it is the case of the common Expected Utility (if, in addition, $\rho = 0$, it is the special case of Expected Value). **c** Comparison with lab experiments: Distributions of Afriat index of our subjects compared with Choi et al.[13]. **d** Validity of MMI. Correlation of Afriat index with Aggregate MMI ($\rho = 0.538$, $p < 0.001$, $n = 33$)

of the aggregate indices (aggregate MMI, Afriat index, and number of GARP violations) and the average change in the BOLD signal in any of the predefined ROIs (Supplementary Table 3).

In addition, we also examined whether similar activations could be found using a nonparametric trial-specific inconsistency index. The same Leave-One-Out procedure was implemented on the number of GARP violations, with no functional form assumptions (henceforth trial-specific violations). At the behavioral level, we found a significant correlation with our

parametric trial-specific MMI index, ($\beta = 0.0009$, $p < 0.0001$, cluster regression). However, the RFX-GLM analysis using nonparametric trial-specific violations index as a regressor did not yield any significant voxels (even with a more liberal threshold, Supplementary Figure 5b). This is likely due to the low variability of the trial-specific violations regressor (73.4% of the data points equal to 0, Supplementary Figure 5a). The Afriat index also yielded a null result (Supplementary Figure 5c). These null results provide additional motivation for using a parametric index with high variability across trials and subjects.

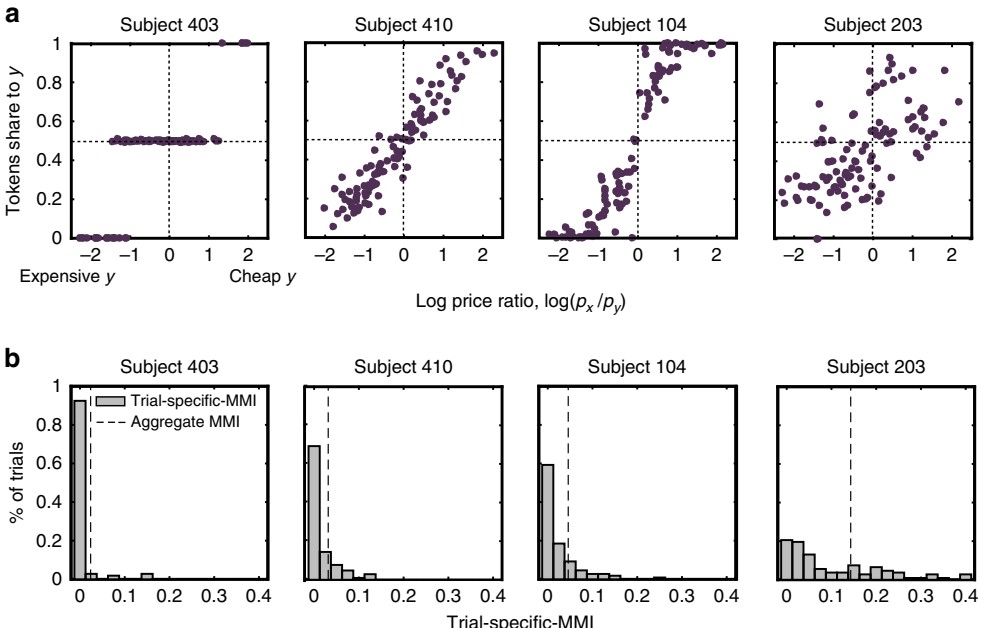

**Fig. 4** Representative subjects. **a** Scatterplots of prominent behaviors: The y-axis represents the share of tokens allocated to the Y account as a function of the log price ratio, $\log(p_x / y)$ (x-axis). As the log price ratio increases, account Y becomes relatively cheaper. Subject 410 equalized expenditures between the two accounts, as she divided tokens proportionally to the price ratio (Cobb–Douglas preferences). Subject 104 exhibited similar behavior, but chose to allocate the entire endowment to the cheaper account in extreme slopes. Subject 403 chose the safe bundle, when the prices of X and Y were relatively similar and allocated most or all her tokens to the cheaper account when the price ratio between the accounts was relatively high (steeper slopes). Even a highly inconsistent subject, like subject 203, was sensitive to changes in prices, with the share of tokens to the Y-account declining as its price rises. **b** Variability of trial-specific MMI: Distributions of the trial-specific MMI for the four representative subjects from panel **a** showcase the heterogeneity of trial-specific MMI scores across and within subjects. For example, subject 203 was highly inconsistent throughout most of her trials, while subject 403 was mostly consistent

**A model of valuation and inconsistent choices**. We now propose a model in which neural variability can generate choice inconsistencies compatible with our empirical findings, including the observation that the BOLD activity correlates positively with both SV and inconsistent choice behavior.

Consider a decision-maker choosing between two alternatives {1, 2} with a valuation for each alternative given by $v_1 > v_2$. Define the first alternative—with the larger valuation—as the consistent choice (i.e., it obeys an ordered utility function $u(\cdot)$, Supplementary Note 1). During a choice, the neural computations which encode and compare valuations are subject to variability. This is represented by a random utility comprised of the valuation $v_i$ plus a random term $e_i$.

$$\tilde{v}_i = v_i + e_i \qquad (1)$$

The alternative with the largest random utility is chosen, therefore the decision-maker might choose inconsistently due to the random component (i.e., $\tilde{v}_2 > \tilde{v}_1$, Fig. 6a). The probability of inconsistent choice is determined by two factors:

1. The distribution of $\tilde{v}_i$.
   A skewed distribution of neural activity is observed in various contexts.[33] Therefore, we assume a skewed distribution for $\tilde{v}_i$, and use the log normal and generalized extreme value distributions as examples.

2. The difference between $v_1$ and $v_2$.
   As the gap between valuations increases, the realization of the random component for the inconsistent alternative must be relatively larger for it to be chosen.

The implications of this model for the neural data are the following: if the BOLD signal correlates with random utility, this signal will be higher when choices are more inconsistent, because

the random component must make up the gap between the higher and lower valuation when the inconsistent option is chosen (Fig. 6a). Because the error distribution is skewed, it is larger on average when the gap in valuations is overcome (Fig. 6b), and particularly so when the gap between the consistent and inconsistent options is larger (Fig. 6c). Therefore, the model predicts that valuation regions of the brain will be more active on inconsistent choices.

The proposed relation between the BOLD signal and choice inconsistency also holds as the number of alternatives increases beyond our binary choice example. Define the choice of alternatives with lower valuations as more inconsistent. When a low value alternative is added to a choice set, its corresponding random component must be larger to overcome the utilities of all higher-valued alternatives. Moreover, the largest realization of an error is required for the lowest value option to be chosen, so on average the largest BOLD signal will arise on a trial in which the lowest valuation alternative is chosen. As the number of low value alternatives increases, this will yield a corresponding increase in the BOLD signal on more inconsistent trials (Fig. 7). This is true whether the BOLD signal correlates with the random utility of the chosen option or the aggregate random utility of all options (Supplementary Figure 8b). Moreover, it is robust over a range of error distributions which have the skewed property consistent with neural activity (Supplementary Figure 8a).

To demonstrate that the random fluctuations in value implied by NRUM are closely related to the observed neural activity, we assessed whether the same correlation pattern between the random utilities and inconsistency index could be observed. Based on the behavior of our subjects, we therefore simulated the valuation process implied by the NRUM and examined the correlation between the random utility valuations (equivalent to

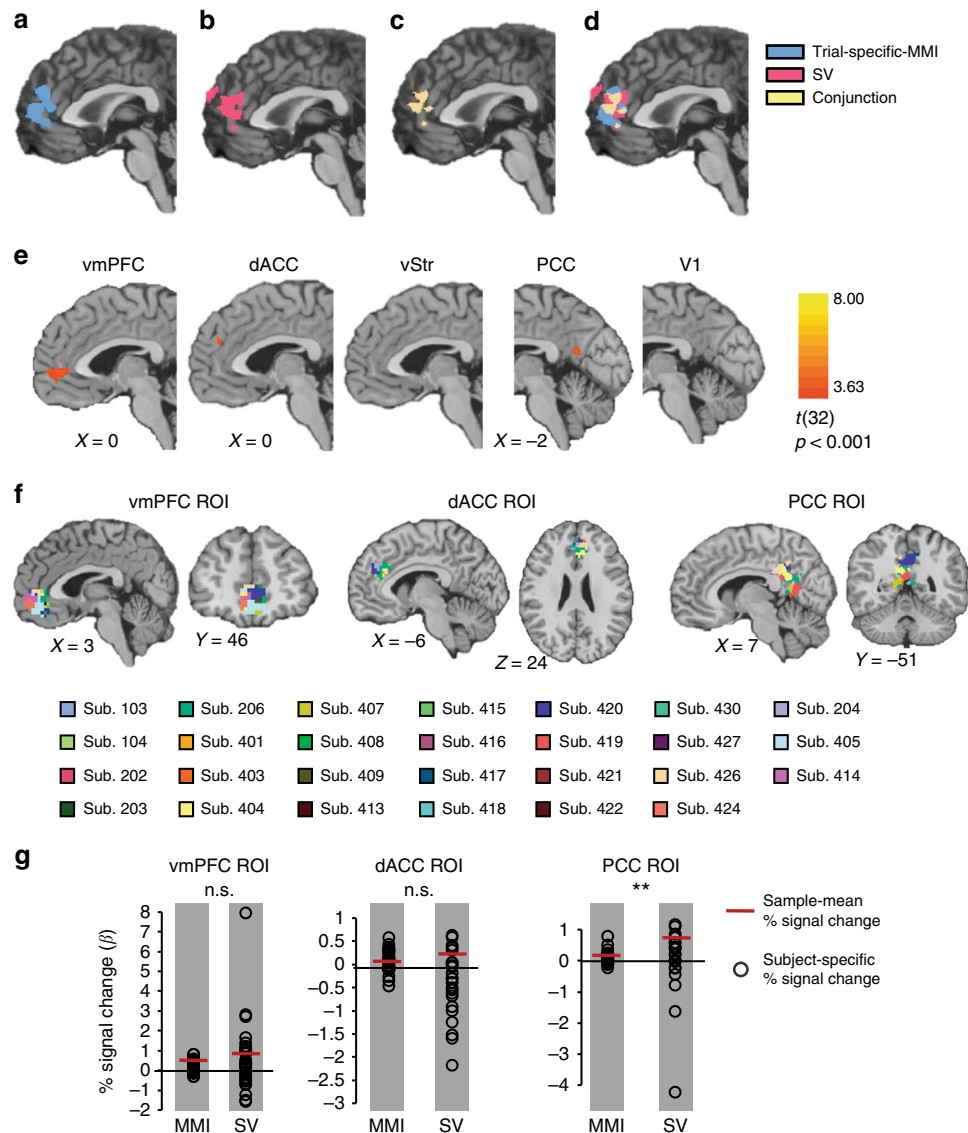

**Fig. 5** Neuroimaging results. **a–d** Whole brain: Results of RFX GLM, $n = 33$, $p < 0.0005$, cluster-size correction, $x = 0$ (MNI coordinates). Model regression: $\mathrm{BOLD} = \beta_0 + \beta_1 \mathrm{RT} + \beta_2 \mathrm{MMI}_{\mathrm{trial\_specific}} + \beta_3 \mathrm{SV} + \beta_4 \mathrm{priceratio} + \beta_5 \mathrm{endowment}$. Six additional motion-correction regressors were included as regressors of no interest. **a** Neural correlates of the trial-specific MMI. **b** Neural correlates of the SV: We present results for the frontal lobe. Other activations are detailed in Supplementary Table 4. **c** Conjunction analysis. **d** Overlay. **e** ROI: The ROI analysis revealed that choice inconsistency was correlated with activation in the dACC, vmPFC ($p$(Bonferroni) < 0.05) and PCC ($p < 0.0005$, cluster-size correction), but neither with vStr nor with V1. RFX GLM, $n = 33$, regression model as in **a–d**. For illustration purposes, we set the threshold to $p < 0.001$ (more stringent than an FDR correction). **f** Subject level: Subject-level analysis representing overlap of the SV and trial-specific MMI in the vmPFC, dACC, and PCC ROIs. For each subject, we conducted a conjunction analysis on the brain areas that significantly tracked trial-specific MMI and SV. Most subjects had an overlap region in the vmPFC (21 of 33), dACC (18 of 33), and PCC (21 of 33). FFX GLM, regression model as in **a–d**. We set a liberal threshold of $p < 0.15$ due to lack of statistical power. For three subjects, the threshold was set to $p < 0.2$, MNI coordinates (see Supplementary Table 5 as well). **g** Comparison of subject-level mean effects (% signal change, $\beta$ values) of trial-specific MMI and SV in the vmPFC, dACC and PCC, using two-sided Wilcoxon sign-rank test (n.s. not significant, **$p < 0.01$). In all panels **a–f**, results are shown on the Colin 152-MNI brain

the BOLD signal) and the inconsistency of their simulated choices. For all subjects, we found significant positive correlation (Table 1 and Supplementary Figure 12), implying that NRUM is consistent with our main empirical finding.

**Dissociation of the SV and trial-specific MMI regressors**. As previously noted, there is orthogonal information in the SV and the trial-specific MMI, since the SV does not depend on the budget set. Indeed, these regressors are weakly negatively correlated ($\beta = -0.00133$, $p < 0.001$, clustered regression by subjects), even if

we control for the expenditure and slope of the budget line. Also, as expected, less than 5% of the variance of the trial-specific MMI is explained by the SV regressor ($R^2 = 0.0496$, see subject-level correlations in Supplementary Table 7). By contrast, there is significant positive correlation between the SV and the BOLD signal, and between the trial-specific MMI and BOLD, when these regressors are included both separately and jointly in the GLM. Therefore, SV and trial-specific MMI appear to be dissociated in our analysis. An orthogonality analysis confirms these results (Supplementary Note 5, Supplementary Figure 11a and b).

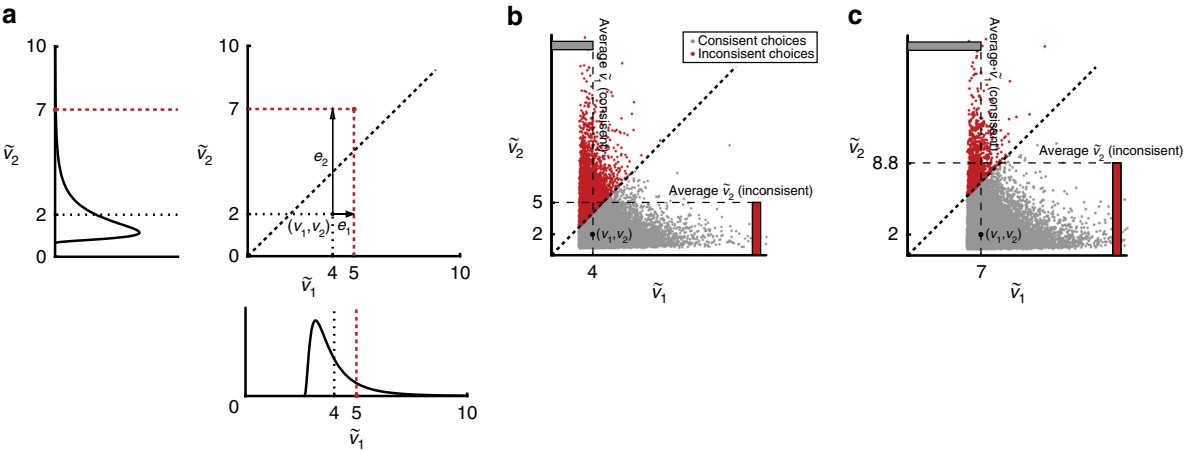

**Fig. 6** The neural random utility model with two alternatives. **a** Inconsistent binary choice: A draw (red) of utilities from a log normal distribution with mean $v_1 = 4$ and $v_2 = 2$, (log s.d. $= 0.7$). In this draw, $\tilde{v}_2 > \tilde{v}_1$, so the inconsistent option (with lower mean valuation) is chosen. **b** Average random utility: A sample of 2,000,000 utilities from the distribution in **a**. The average utility is higher when the inconsistent option is chosen. **c** Larger difference: A sample of utilities from distributions with a larger difference in mean valuations (log normal, means $v_1 = 7$ and $v_2 = 2$, (log s.d. $= 0.7$). The average utility of an inconsistent choice is larger when the choice is *more* inconsistent (the difference in mean valuations is larger compared to **b**)

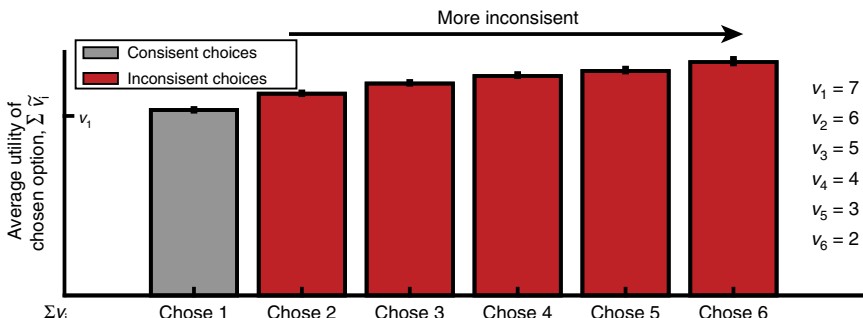

**Fig. 7** The neural random utility model with six alternatives. The alternatives are ranked in value ($v_1 = 7,...,v_6 = 2$) with the highest valued alternative termed the consistent choice. For inconsistent choices, the average utility of the chosen option increases as worse alternatives are chosen. This is because the random term, $e_i$, had to be much larger (e.g., compare Chose 6 vs. Chose 2). Error bars indicate standard errors

| **Table 1 Simulation results** | | | | | |
|---|---|---|---|---|---|
| **SID** | **Gumbel dist.** | **Log normal dist.** | **SID** | **Gumbel dist.** | **Log normal dist.** |
| 103 | 0.1245* | 0.1484* | 412 | 0.1084* | 0.1105* |
| 104 | 0.1345* | 0.1568* | 413 | 0.1201* | 0.1089* |
| 202 | 0.1048* | 0.1341* | 414 | 0.1273* | 0.1662* |
| 203 | 0.1233* | 0.1261* | 415 | 0.1227* | 0.1175* |
| 204 | 0.1290* | 0.1581* | 416 | 0.0543* | 0.0455* |
| 205 | 0.1266* | 0.1288* | 417 | 0.1084* | 0.1391* |
| 206 | 0.1345* | 0.1369* | 418 | 0.1312* | 0.1500* |
| 401 | 0.1244* | 0.1343* | 419 | 0.0934* | 0.0916* |
| 402 | 0.1002* | 0.0914* | 420 | 0.1120* | 0.1339* |
| 403 | 0.0963* | 0.1139* | 421 | 0.1303* | 0.1301* |
| 404 | 0.0607* | 0.0706* | 422 | 0.0779* | 0.0701* |
| 405 | 0.1148* | 0.1143* | 424 | 0.1053* | 0.0889* |
| 406 | 0.1196* | 0.1187* | 426 | 0.1240* | 0.1259* |
| 407 | 0.1289* | 0.1620* | 427 | 0.1200* | 0.1133* |
| 408 | 0.1208* | 0.1298* | 428 | 0.1281* | 0.1746* |
| 409 | 0.1227* | 0.1484* | 430 | 0.1365* | 0.1563* |
| 410 | 0.1206* | 0.1207* | | | |

The correlation coefficients $r$ for the pooled simulated series per subject using two skewed distributions for the random fluctuations in value, the Gumbel and log normal (*$p < 10^{-10}$).

Finally, a psychophysical interaction (PPI) analysis between our predefined ROIs (vmPFC, dACC, and PCC) and other brain regions revealed that the activity in the seed regions and other brain areas interacted with the SV and trial-specific MMI regressors (Methods and Supplementary Figure 11c). As might be expected, all three seed regions were interacting with motor and visual regions, for reasons likely related to task execution. In addition, we found interaction with other value-related regions, such as the Insula and dlPFC[23,42]. Importantly, though, only under the SV context did we find interactions between the seed regions—specifically, the PCC and vmPFC interact with the dACC. This may indicate that choice inconsistency, represented by trial-specific MMI, results from a spontaneous or random process, not coordinated across the different nodes of the "value network".

**Controlling for other sources of choice inconsistency**. It is possible that inconsistency arises due to other sources of noise—like imprecision in motor execution or the numerical representation of the choice options. To control for these alternative explanations, we conducted additional analyses using two functional localizers that were collected at the end of the main experiment.

First, in a motor imprecision localizer task, subjects were asked to reach a predefined location marked as a circle on the line (Supplementary Figure 1a). Motor imprecision was measured by the average Euclidean distance between the predefined target and the actual location the subject chose. Across subjects, the average motor noise and the Aggregate MMI were not significantly correlated ($r = 0.144$, $p = 0.511$, Supplementary Figure 1c). In addition, as might be expected, frontal lobe activity in premotor and motor areas (peak voxel at [17, 25, 42], MNI coordinates), positively correlated with the imprecision regressor (Supplementary Figure 1d). However, no voxels conjointly represented both motor imprecision and inconsistency level (i.e., trial-specific MMI).

In a second, numerical imprecision localizer task (Supplementary Figure 1b), we estimated the numerical execution of each subject in reaching a predefined $\{X,Y\}$ coordinate on the line. Numerical imprecision was measured by the average Euclidean distance between the coordinates and the actual location subjects chose. The average numerical execution imprecision per subject was not correlated with the Aggregate MMI ($r = 0.105$, $p = 0.61$, Supplementary Figure 1f). We also did not find any neural correlates with the imprecision of numerical execution (Supplementary Figure 1g). These analyses suggest that the neural activations of inconsistency are not due to imprecisions in motor or numerical execution, and are mainly observed in value-related brain areas (Supplementary Figure 1e).

**Controlling for choice difficulty.** In binary choice tasks, choice difficulty is usually considered to be the difference in the SVs between the two options ($\Delta$SV)—the smaller $\Delta$SV, the more similar the options are, and the higher the difficulty level[44,46–48]. Similar intuition holds for continuous choice sets, though a measure of difficulty must account for both the subject's preferences and the larger number of choices. To address the role of choice difficulty in our dataset, and its possible relationship to choice inconsistency, we propose an index of choice difficulty in continuous choice sets and include it as a control in our main GLM (see the Methods section for details).

In our Choice Simplicity index, the closer its value is to 0, the more difficult is the choice. As expected, difficult choices lead to longer RTs ($\beta = -2.782$, $p < 0.0001$, clustered regression by subjects). Trial-specific MMI is negatively correlated with our Choice Simplicity index ($\beta = -0.0302$, $p < 0.0001$), meaning, the more difficult is the choice problem, the higher is the corresponding inconsistency level. However, the index also explains little of the variance in trial-specific MMI scores ($R^2 = 0.0137$), therefore, the relationship between choice difficulty and inconsistency is weak.

It is important to note that the random utility model that we propose predicts these findings. As noted in our modeling, choice difficulty is a key determinant of the probability of inconsistent choice; we would expect that choices are more inconsistent on more difficult problems. However, if there was no variability in valuation, then even the most difficult choice would not lead to inconsistencies (i.e., choice difficulty alone cannot lead to choice inconsistency). It is precisely the variability that "connects" choice difficulty to inconsistency, with the obvious implication that more difficult problems are more likely to produce inconsistency. Therefore, a weak negative correlation between the Choice Simplicity index and the level of inconsistency is expected.

In the RFX GLM analysis, our main findings for trial-specific MMI and SV clusters hold when controlling for difficulty (Supplementary Figure 9a–f). Moreover, the Choice Simplicity index was correlated with ACC activation (among other brain regions, but not the vmPFC and PCC), as extensively suggested

by the literature[44,49] ($p < 0.0005$, cluster-size corrected, Supplementary Figure 9g).

**Controlling for the role of confidence in decision-making.** Another possible source for choice inconsistency is low levels of confidence in one's own choice. Following Lebreton et al.[50], we modeled levels of confidence as the second-order polynomial of SV. Similarly to Lebreton et al.[50], we find a quadric relationship between RT and confidence levels, when controlling for the first-order polynomial of SV ($\beta = -0.00155$, $p < 0.005$), indicating subjects had the longest RTs in intermediate confidence-level choices.

As expected from the dissociation analysis above, we found that trial-specific MMI was correlated with our measure for confidence ($\beta = -0.000027$, $p < 0.0001$ in a clustered regression by subjects). Such result indicates that low levels of confidence correlate with high inconsistency scores; however, the $R^2$ of the model is very low (0.0395). When we added the confidence predictor to the RFX GLM, our main results hold, suggesting that the BOLD activity in mPFC and ACC was larger on inconsistent trials, even after controlling for confidence (Supplementary Figure 10).

**Robustness of the trial-specific MMI.** We ensured our results remained robust even after controlling for changes in heuristics over the blocks of the experiment (Supplementary Note 3, Supplementary Figure 4 and Supplementary Tables 1 and 2). The results remain unchanged also when we control for mis-specification of the utility function, by using a different functional form (Methods, Supplementary Note 4 and Supplementary Figure 6).

## Discussion

In this study, we explored the neural computations that give rise to inconsistent choice behavior when the framing and context of the choice problems are stable. We introduced a novel trial-specific inconsistency index and found that it was positively related to activations in the vmPFC, dACC, and PCC which, strikingly, lie in the same regions of cortex as value representations. Moreover, the functional connectivity networks of the SV are more interconnected than for inconsistency, suggesting that inconsistent choices might be driven by idiosyncratic fluctuations within these regions. The main results were corroborated with an ROI analysis on anatomically defined brain regions, and were robust to several alternative explanations including the influence of motor or numerical noise. We also proposed a novel index for measuring choice difficulty on a continuous budget line, and demonstrated our main result is robust to difficulty. Finally, including a proxy for confidence[50] does not alter our main findings.

Our main empirical finding is a positive correlation between the severity of choice inconsistency and the BOLD signal located primarily in value-related areas. Based on our behavioral and neural data, we proposed a computational model in which inconsistent choice behavior originates from the variability in neuronal value computation. In cases where neural variability is large enough to overcome the value difference between the alternatives, choices of low valuation alternatives may occur. If the error distribution is skewed, we should expect to see higher neural activation in value regions for more inconsistent choices. Therefore, this study provides evidence consistent with the view that choice inconsistency arises from variability in regions of the human brain that are known to be responsible for value computation.

The hypothesis that choice inconsistency is tied to variability in valuation is not novel[18]. However, the standard explanation in

economics is that this variability arises from limitations in the data available to the researcher, not that choice itself is stochastic[51]. More recently, decision theorists have proposed that the source of choice inconsistency might be more fundamental; that is, choice is stochastic because utilities are stochastic[18,19,52,53]. While such theories place important empirical constraints on the pattern of inconsistent behavior, these constraints are weak because utilities are assumed to be unobservable. By contrast, the ability to observe the neural valuations of choice alternatives on a given trial enables a much stricter test of the hypothesis that choice inconsistency is due to stochastic valuations[20]. Therefore, we should expect to see the empirical results reported here if inconsistent choice behavior arises from stochastic valuations computed in value regions of the brain.

The main methodological contribution of this study is the trial-level index of inconsistency. Existing inconsistency indices are aggregate measures, which use an entire subject's dataset to provide one inconsistency score. For this reason, aggregate measures which correlate BOLD signal across subjects lose statistical power; they ignore trial-by-trial variations in behavior—and its neural foundations—thus cannot take advantage of the rich trial-level measurements provided by the MRI scanner. In particular, the most informative trials that induced inconsistent choices are lost when averaging over all trials. By contrast, our proposed index tracks trial-by-trial variations in behavior and neural activations, therefore provides insight into the valuation and choice process when a subject chooses inconsistently. We should note its use need not be limited to neuroeconomic studies; standard behavioral laboratory experiments can use the trial-specific MMI index to test theories which imply varying behavior across trials (e.g., choice dynamics).

At first glance, our empirical results might seem surprising given both lesion and stimulation studies which demonstrate that activity in value-related regions is necessary for consistent choice[37,54]. Moreover, Polania et al.[54] find that choice behavior becomes less accurate when frontal–parietal coupling is disrupted by tACS. However, the results from these studies are compatible with our proposed computational model for choice behavior. When value-related regions are absent or disrupted, choices are highly inconsistent because of the limited ability of the brain to compute the valuations necessary for consistent choice behavior. When these regions are intact, value signals can be computed, but with a degree of variability inherent to neural computation. Thus, choices are largely consistent, but exhibit a pattern in which inconsistent choices correlate with an increase in aggregate activity. As further evidence, lesioned subjects in Camille et al.[37] had an Afriat Index of 0.1 on average, compared with an average of 0.0623 in our sample, even though our subjects were facing a much more difficult task (11 trials vs. 108 trials).

The neural random utility model represents a parsimonious account of the variability in value signals during a decision[20,29]. As such, the trial-by-trial variability we propose can arise from multiple sources, including higher order cognitive processes such as fluctuations in attention or heuristics or lower-level process like neuronal noise. At a computational level, a number of previous studies have explored the role of noisy computations in choice behavior, typically in the form of a bounded accumulation model, in which a noisy decision signal accumulates to some threshold (the drift-diffusion model, DDM)[30]. These models have found support in both single-neuron recordings[30] and human imaging studies[55], with the fMRI studies in particular observing value signals in the vmPFC. Since the NRUM is a general formulation of bounded accumulation models[29], this large set of accumulation models can provide a computational account of the results we observe. Of primary importance, the distributions used in our examples (either Log normal or Gumbel distributions[18])

are skewed with a long right tail, a property consistent with neural data at both the level of single-neuron firing rates and aggregate network-level measures[33]. Therefore, our results are compatible with the evidence for a skewed distribution of neural activity in value-related regions. To demonstrate that the valuations implied by the NRUM are closely related to the observed neural activity, we simulated the valuation process of our subjects based on their observed behavior and demonstrated a correlation between their random utilities (equivalent to the BOLD signal) and the inconsistency of their simulated choices (equivalent to the trial-specific MMI index).

What might be the source of this variability in value computation, and why should it persist in decision-making? Variability is inherent to neural computation[56], arising from thermodynamic noise at the cellular and synaptic level, and is present at all stages from primary sensory systems to motor execution[56,57]. Network computations can filter, or integrate, this noise, but the maximal signal–noise ratio is bounded due to physiological constraints[58,59]. Thus, some constrained optimal degree of noise persists at the network level across domains from perceptual[30] to value-based choice[28,31,32]. In a seminal article, Fox et al.[60] find that "inconsistency in perception or performance should not be automatically attributed to fluctuations in task-related cognitive processes such as attention, but could also be due to ongoing fluctuations in intrinsic neuronal activity". Indeed, a recent study conducted to explicitly separate the sources of behavioral variability finds that 89% of deviations from optimal choice can be attributed to errors in value inference, rather than sensory processing or action selection[61]. This is consistent with our finding of a limited role for noise in motor output or numerical representation in generating inconsistent choices. Instead, the deviations from consistency were observed in valuation regions, suggesting that the value of choice options might be fluctuating on a trial-by-trial basis. This result is in line with non-human studies: In primates, variability in the firing rate of orbitofrontal cortex neurons predicted choices of near indifferent alternatives[62], while stability in neural populations in the medial frontal cortex accounted for variability in choice in rats[63]. Furthermore, neural variability can be influenced by varying levels of attention[64–66]. These results point to a speed-accuracy tradeoff in decision-making governed by metabolic costs[67].

Taken together, these results question whether the form of inconsistency we observe should be considered sub-optimal. Our results are consistent with a degree of constrained-optimal variability around the normatively defined benchmark of a utility representation. In fact, our results may be interpreted as implying that inconsistent choice behavior is an integral feature of human decision-making.

## Methods

**Participants.** Thirty-eight subjects participated in the study (17 females, mean age 25.3, 18–36). Subjects gave informed written consent before participating in the study, which was approved by the local ethics committee at Tel Aviv University and by the Helsinki Committee of Sheba Medical Center. Three subjects were dropped due to sharp head movements (> 3 mm). Another subject opted out from the experiment before completing the scan, and another subject was dropped due to anatomical abnormalities. We therefore report the data for the remaining 33 subjects.

**Experimental task.** We used a modification of the task presented by Choi et al.[13]. On each trial, subjects faced a visualization of a budget line (Fig. 2a). Each discrete $(x, y)$ point on the budget line corresponds to a lottery with a 50% chance of winning the tokens allocated to account $X$ (the $x$-axis coordinate) and a 50% chance of winning the tokens allocated to account $Y$ (the $y$-axis coordinate). Thus, the budget line describes the possible allocations to accounts $X$ and $Y$ on a two-dimensional graph. On each trial, the subject was asked to choose the desired bundle (of $X$ and $Y$ tokens) from the budget line, knowing that only one of the accounts will be realized. At the end of the experiment, one of the trials was randomly selected for monetary payment (to satisfy incentive compatibility).

The subject won the monetary value of the tokens allocated to the winning account on the trial drawn for the payment. Each token was worth 5 NIS ($1 ≅ 3.5 NIS).

In this task, the slope of the budget line determines the price of one unit of account $X$ relative to one unit of account $Y$. We varied and randomized the budget lines (slopes and endowments) across trials and subjects. The x-axis and y-axis were scaled from 0 to 100 tokens, and the resolution of the budget line was 0.1 tokens. Subjects could not choose inside the budget line. In trials where subjects did not make any choice in the allotted time, a text reading "No choice was made" appeared on the screen. These trials were excluded from the analysis (35 trials out of 3,564 total trials). The average prize was 191.6 NIS + 100 NIS show up fee.

**fMRI session**. Subjects performed the experimental task using an fMRI compatible trackball to choose their preferred bundle. On each trial, subjects had a maximum of 12 s to make their choices, followed by a 9 s variable inter-trial-interval (jittered between trials). If subjects made their choice before the end of the maximal 12 s, the remaining time was added to the ITI. There were 27 trials in each block, and each subject completed four blocks, for a total of 108 trials.

After completing the main task, we obtained an anatomical scan and two functional localizers, one numerical and one motor (counter-balanced), aimed to control for alternative sources of choice inconsistency (see Functional Localizers below).

**Instructions and pre-scan practice**. Before the scan, subjects read an instruction sheet and completed a pre-scan questionnaire to verify the task is clear. The instructions included many examples and were written in simple terms to avoid confusion. In the pre-scan questionnaire, subjects were given several decision problems (i.e., graphs representing different budget sets) and were asked to identify intersections with the axes, identify the cheaper account, and calculate the possible winning prize (in terms of both tokens and NIS) for a specific (x, y) coordinate. After the pre-scan questionnaire, the experimenter went over their answers. In case the subject made a mistake, the experimenter explained the instructions orally, and then repeated the question in the questionnaire until the subject answered correctly. See an English translation of the instructions and pre-scan questionnaire in Supplementary Notes 6 and 7. Thereafter, subjects completed a practice block in front of a computer, using a similar trackball to the one used inside the fMRI, in order to imitate the motor movements required during the scan. The budget sets in the practice block were predefined to ensure all subjects encountered the same (substantial) variation of slopes and endowments.

**Image acquisition**. Scanning was performed at the Strauss Neuroimaging Center at Tel Aviv University, using a 3 T Siemens Prisma scanner with a 64-channel Siemens head coil. To measure blood oxygen level-dependent (BOLD) changes in brain activity during the experimental task, a T2*-weighted functional multi-band EPI pulse sequence was used (TR = 1.5 s; TE = 30 ms; flip angle = 70° matrix = 86 × 86; field of view (FOV) = 215 mm; slice thickness = 2.5 mm; band factor = 2). Fifty-two slices with no inter-slice gap were acquired in ascending interleaved order, and aligned 30° to the AC–PC plane to reduce signal dropout in the orbitofrontal area. Anatomical images were acquired using 1-mm isotropic MPRAGE scan, which was comprised from 208 axial slices without gaps at an orientation of −30° to the AC–PC plane.

**fMRI data preprocessing**. BrainVoyager QX (Brain Innovation) was used for image analysis, with additional analyses performed in MATLAB (MathWorks). Functional images were sinc-interpolated in time to adjust for staggered slice acquisition, corrected for any head movement by realigning all volumes to the first volume of the scanning session using six-parameter rigid body transformations. Spatial smoothing with a 6-mm FWHM Gaussian kernel was applied to the fMRI images. Images were then co-registered with each subject's high-resolution anatomical scan and normalized using the Montreal Neurological Institute (MNI) template. All spatial transformations of the functional data used trilinear interpolation.

**The General Axiom of Revealed Preference (GARP)**. Consider a finite dataset $\mathbf{D} = \{(\mathbf{p}^i, \mathbf{x}^i)\}_{i=1}^n$, where $\mathbf{x}^i \in \mathbb{R}_+^k$ is the subject's chosen bundle at prices $\mathbf{p}^i \in \mathbb{R}_{++}^k$ (k is the number of goods in the bundle). Bundle $\mathbf{x}^i$ is

1. Directly revealed preferred to another bundle $\mathbf{x}$, denoted $\mathbf{x}^i R^0 \mathbf{x}$, if $\mathbf{p}^i \mathbf{x}^i \geq \mathbf{p}^i \mathbf{x}$.
2. Strictly directly revealed preferred to bundle $\mathbf{x}$, denoted $\mathbf{x}^i P^0 \mathbf{x}$, if $\mathbf{p}^i \mathbf{x}^i > \mathbf{p}^i \mathbf{x}$.
3. Revealed preferred to bundle $\mathbf{x}$, denoted $\mathbf{x}^i R \mathbf{x}$, if there exists a sequence of observed bundles $(\mathbf{x}^j, \mathbf{x}^k, \ldots, \mathbf{x}^m)$, that are directly revealed preferred to one another, $\mathbf{x}^i R^0 \mathbf{x}^j, \mathbf{x}^j R^0 \mathbf{x}^k, \ldots, \mathbf{x}^m R^0 \mathbf{x}$. Relation $R$ is therefore the transitive closure of the directly revealed preferred relation.

**D** satisfies the General Axiom of Revealed Preference (GARP), if every pair of observed bundles, $\mathbf{x}^i R \mathbf{x}^j$, implies $\neg(\mathbf{x}^j P^0 \mathbf{x}^i)$. We say a subject is consistent *iff* she satisfies GARP. We say a utility function $u(\mathbf{x})$ rationalizes **D** if $\mathbf{x}^i R^0 \mathbf{x}$ implies $u(\mathbf{x}^i) \geq u(\mathbf{x})$. According to the Afriat theorem[1,2], there exists a well-behaved utility function (continuous, monotone, and concave) that rationalizes the data *iff* the subject satisfies GARP (Supplementary Figure 2b). Otherwise, a strict cycle of

choices exists and we say that **D** violates GARP. By the Afriat's theorem, if the dataset **D** does not satisfy GARP, then the subject cannot be described as a non-satiated utility maximizer and is therefore said to be inconsistent[1,2].

**Aggregate inconsistency indices**. As subjects often violate GARP, and therefore are inconsistent[11,14,45], one would like to measure their level of inconsistency. The simplest way would be to count the number of GARP violations. Other well-known nonparametric inconsistency indices are Afriat index[2,34], Varian index[35] and Houtman–Maks index[36] (see Supplementary Note 1 for detailed descriptions of these indices). For each subject, we calculated the number of GARP violations and Afriat index for the entire experiment (108 trials). We were unable to compute Varian index and Houtman–Maks index at the aggregate level as they are hard computationally[35] (see Appendix B in Halevy et al.[45]). In the current study, we also compute a parametric index for inconsistency—the Aggregate MMI.

**Aggregate MMI**. Following Halevy et al.[45] consider the continuous and non-satiated utility function $u(\cdot)$ as representing the preferences of the subject. $u(\cdot)$ induces a complete ranking on the bundles such that if $u(\mathbf{x}) > u(\mathbf{y})$, then bundle $\mathbf{x}$ is preferred to bundle $\mathbf{y}$. In addition, each actual choice induces a partial order on the bundles since when a subject chooses bundle $\mathbf{x}^i$, she, by the principle of revealed preference, ranks this bundle over all other feasible bundles. If these two rankings are compatible for every choice made by the subject, $u(\cdot)$ rationalizes **D**. Otherwise, if these two rankings are incompatible for some choice according to $u(\cdot)$, some feasible bundles are ranked higher by $u(\cdot)$ than the chosen Bundle $\mathbf{x}^i$. For every observation, the incompatibility between the two rankings can be measured by the minimal expenditure (parallel inward movement of the budget line), such that the adjusted budget set does not include any bundle that is strictly preferred over $\mathbf{x}^i$ according to $u(\cdot)$. Halevy et al.[45] show that this measure is exactly the well-known money metric[68]. Formally, given the prices $\mathbf{p}^i$, the money metric $m(\mathbf{x}^i, \mathbf{p}^i, u)$ for observation $i$ is the minimal expenditure required for the dataset to include a bundle $\mathbf{y}$ such that $u(\mathbf{y}) \geq u(\mathbf{x}^i)$:

$$m(\mathbf{x}^i, \mathbf{p}^i, u) = \min_{u(\mathbf{y}) \geq u(\mathbf{x}^i)} \mathbf{p}^i \mathbf{y} \qquad (2)$$

We normalize the money metric measure by the original expenditure, and therefore the adjustment for trial $i$ is $v_i^*(\mathbf{D}, u) = 1 - \frac{m(\mathbf{x}^i, \mathbf{p}^i, u)}{\mathbf{p}^i \mathbf{x}^i}$ (Fig. 1a). Hence, if no adjustment is needed, $v_i^*(\mathbf{D}, u) = 0$. Next, we aggregate the adjustments for all observations using some aggregator function $f(v^*(\mathbf{D}, u))$ (specifically, the average sum of squares), and get a measure of the incompatibility between the utility function $u(\cdot)$ and the dataset **D**, given the aggregator $f$. Finally, we iterate over all utility functions in the set of utility functions $\mathcal{U}$ under investigation and look for the one with the smallest incompatibility with the dataset **D**. The MMI, denoted $I_M(\mathbf{D}, f, \mathcal{U})$, interprets the incompatibility between this utility function and **D**, as the incompatibility between the set of utility functions $\mathcal{U}$ and **D** given the aggregator $f$.

$$I_M(\mathbf{D}, f, \mathcal{U}) = \inf_{u \in \mathcal{U}} f(v^*(\mathbf{D}, u)) \qquad (3)$$

Halevy et al.[45] prove that had we examined the set of all continuous non-satiated utility functions (denoted $\mathcal{U}^C$), the MMI would be equal to the nonparametric Varian inconsistency index, denoted $I_v(\mathbf{D}, f)$[35]. As it is not feasible to examine all utility functions in this set, they propose restricting to a specific functional form. The MMI thus includes a misspecification element. Fortunately, the MMI is separable additive in Varian inconsistency index and the misspecification:

$$I_M(\mathbf{D}, f, \mathcal{U}) = I_v(\mathbf{D}, f) + \text{Misspecification} \qquad (4)$$

The computation of MMI yields two measures: (a) the computation of aggregate MMI; (b) elicitation of subject-specific utility function parameters. The subject-specific utility function is the function for which aggregate MMI is minimal, and hence constitutes the best fit for the subject's choices among the investigated family of utility functions $\mathcal{U}$.

**Trial-specific index**. A Leave-One-Out procedure. Let $\varepsilon_D$ be an aggregate inconsistency index of dataset **D**. Let $\mathbf{D}_{-i}$ be a subset of **D** that includes all $n-1$ trials but the $i^{\text{th}}$ observation. Let $\varepsilon_{D_{-i}}$ be the aggregate inconsistency index of $\mathbf{D}_{-i}$, and let $\left(\varepsilon_D - \varepsilon_{D_{-i}}\right)$ be the trial-specific inconsistency index of trial $i$.

The best practice would be to use Varian index as $\varepsilon_D$, as it is the nonparametric index with the highest number of degrees of freedom. However, it is not possible to compute Varian index for datasets with 108 observations in feasible time. Therefore, we had to choose between two other alternatives. The first was to use computational-convenient nonparametric indices (Afriat index and GARP violations). Nevertheless, when using those nonparametric indices, the trial-specific index $\left(\varepsilon_D - \varepsilon_{D_{-i}}\right)$ usually equals 0 and therefore lacks the required variability across trials. The distribution of a nonparametric trial-specific index is depicted in Supplementary Figure 5a. One should notice that the variability in the trial-specific index is important when using the General Linear Model (GLM) to correlate the BOLD signal, as otherwise it lacks statistical power. Therefore, we picked aggregate MMI as $\varepsilon_D$, and refer to $\left(\varepsilon_D - \varepsilon_{D_{-i}}\right)$ as trial-specific MMI. The code-package used

to compute aggregate MMI and trial-specific MMI is available as open source in https://github.com/persitzd/RP-Toolkit.

**Parametric utility**. For parametric family of utility functions, we use the Disappointment Aversion model with CRRA functional form[69], as it includes many well-known types of preferences in the context of risk[13,45] (see also Appendix D of Halevy et al.[45]). Formally,

$$SV(x_1^i, x_2^i) = \gamma\omega(\max\{x_1^i, x_2^i\}) + (1 - \gamma)\omega(\min\{x_1^i, x_2^i\}), (DA) \quad (5)$$

$$\gamma = \frac{1}{2 + \beta}, -1 \leq \beta < \infty \quad (6)$$

$$\omega(z) = \begin{cases} \frac{z^{1-\rho}}{1-\rho}, \rho \geq 0 \\ \ln(z), \rho = 1 \end{cases} (CRRA) \quad (7)$$

where $\gamma$ is the weight of the better outcome, and $\omega$ is a CRRA utility index with a relative risk aversion parameter $\rho$. When $\beta = 0$, this is the common Expected Utility function with parameter $\rho$ (if, in addition $\rho = 0$, it is Expected Value and when $\rho = 1$ it is the Cobb–Douglas with equal exponents). When $\beta > 0$ the individual over-weights the probability that the lottery will yield the lower prize ("disappointment aversion"). When $\beta \to \infty$, the subject cares only about the element with the lower quantity and therefore her optimal behavior would be to always choose the safe bundle, so that the lottery is meaningless (i.e., Leontief preferences). When $\beta < 0$, the individual overweights the probability that the lottery will yield the higher prize ("elation seeking"). When $\beta = -1$, the subject cares only about the element with the larger quantity (see Supplementary Figure 3b).

To use SV as a parametric regressor in our analysis, we calculated the value of the Disappointment Aversion model with CRRA functional form at the chosen bundle $(x_1, x_2)$ in each trial $i$, using the subject's recovered parameters (using the MMI), $\beta$ and $\rho$.

**Assessing the NRUM and inconsistency**. On each trial, we reconstructed the set of bundles each subject encountered and calculated the SV using the parameters elicited by MMI method (Fig. 3c). These correspond to the $v_i$s in the proposed model.

We calibrated two skewed distributions (the zero-mode Gumbel distribution and the zero-mean log normal distribution) for the neural noise $e_i$ using the observed inconsistency level of each subject. The standard deviation of the distributions was chosen so that the average level of the Afriat inconsistency index matched the observed index.

Based on the $v_i$s and the calibrated distributions for $e_i$, we calculated the random utility values ($\tilde{v}_i$) for each alternative in every trial. For each trial, following value maximization, the chosen bundle was the alternative with the highest random utility value. We repeated this procedure for each subject 1,000 times for each distribution. Hence, we obtained 1,000 simulated datasets for each subject (for each of the two noise distributions).

Next, we tested whether the simulated datasets are compatible with our interpretation of the neural results. Note that we cannot simply use the trial-specific MMI index here, because we have already based the simulation on the parameters elicited by MMI to calculate the $v_i$s. This would amount to double-dipping the data. Instead, we used the trial-specific Afriat index as a proxy for the trial-specific MMI index (note they are highly correlated, Fig. 3e). For each simulated trial, we calculated the noise of the chosen bundle as a proxy for the valuation noise in the BOLD signal. We pooled these two series across simulations and trials, and then assessed their correlation.

**Whole-brain analysis of choice inconsistency**. To identify the neural correlates of choice inconsistency, we estimated a general linear model (GLM) with 11 predictors. The trial-by-trial inconsistency index trial-specific MMI and the trial-by-trial SV, entered for the total trial duration up until the subject made a choice, normalized and convolved with the canonical hemodynamic response function (HRF). We modeled RT using a boxcar epoch function, whose duration was equal to the RT of the trial[70]. The other predictors included the price ratio of the budget set (the slope), and the endowment measured by the safe portfolio on the 45 degrees line from the origin.

All these predictors were entered for the trial duration, normalized and convolved with the HRF. In addition, six motion-correction parameters and the constant were included as regressors of no interest to account for motion-related artifacts.

**ROI analysis of choice inconsistency**. We also conducted a region of interest (ROI) analysis, in order to increase the power of the statistical test. We defined the vmPFC and vStr ROIs based on the masks provided by Bartra et al.[21]. For the dACC, we drew a 12-mm sphere around the peak voxel that Kolling et al.[43] reported. For the PCC and V1 ROIs, we used neurosynth.org meta-analyses masks. We then conducted the same RFX GLM reported above and correlated trial-specific MMI and SV with BOLD activity extracted from each of the ROIs.

**PPI analysis**. The time series of the BOLD signal in each ROI was z-scored to generate the time series of the neuronal signal for each source region as the physiological variable in the PPI. We tested (separately) each parametric regressor, SV and trial-specific MMI, as the psychological variable, suggesting that a given region can be connected to distinct regions/networks depending on task context[71]. The psychological regressors were normalized and convolved with the canonical HRF and entered to the regression model. An additional regressor represented the interaction between the psychological and physiological factors, and indicated in which areas there were significant differential functional connectivity with each seed ROI. We modeled RT similarly to the whole-brain analysis, and used the price ratio of the budget set and the endowment as control predictors. These predictors were entered for the trial duration, normalized and convolved with the HRF as well. In addition, six motion-correction parameters and the constant were included as regressors of no interest to account for motion-related artifacts.

We used RFX for the group-level analysis, and set the threshold to $p < 0.0005$ with cluster-size correction, similarly to the main GLM analysis. We hence ran six models in total (3 ROIs × 2 psychological contexts).

**Orthogonality analysis**. To examine the neural footprints of the non-correlated part of the SV and trial-specific MMI predictors, we used an orthogonality analysis. We ran a clustered regression of trial-specific MMI on SV and obtained residuals $\tilde{e}$, and similarly obtained residuals $\tilde{u}$ from a clustered regression of SV on trial-specific MMI. We repeated the RFX-GLM as in the whole-brain analysis, but replaced trial-specific MMI with $\tilde{e}$, and similarly ran the RFX-GLM replacing SV with $\tilde{u}$.

**Measuring choice difficulty**. Since each decision problem in our task is continuous, we cannot simply use $\Delta SV$ as a choice difficulty index. Moreover, choice difficulty in our task is determined by the subject's own preferences and not only by the slopes of the budget sets. For example, a risk-averse subject might struggle with steep slopes, as the temptation for allocating all tokens to one account rises. A risk-seeking subject, on the other hand, will have difficulties in moderate slopes. Therefore, we derive a novel measure for choice difficulty, which considers both subject's elicited parameters and the continuum of the budget set.

We discretized each budget line into 1000 possible bundles. We then calculated the subjective value $v_{i,b,s}$ of each bundle $i$ along each budget set $b$ for each subject $s$ using the parameters elicited by the MMI. Next, we subtracted $v_{i,b,s}$ from the maximal subjective value of that budget line to obtain a $\Delta V_{i,b,s}$ measure for each bundle along the budget set (analogous to taking the difference between two options in a binary choice design). We then averaged all the $\Delta V_{i,b,s}$ across the 1000 bundles, and normalized the result by the endowment of the given budget set to be able to compare across trials. Formally, denote $V_{b,s} = max_{i\in[1,\dots,1,000]} v_{i,b,s}$. Then:

$$\text{Choice Simplicity}_{b,s} = \frac{\frac{\sum_{i=1}^{N}[V_{b,s} - v_{i,b,s}]}{N}}{\text{Endowment}_b} \quad (8)$$

where $i\in [1,\dots,1000]$ is the bundle ($N = 1000$); $b\in [1,\dots,108]$ is the decision problems and $s\in [1,\dots,33]$ is the subject.

Difficult choices occur when the bundles along the budget line have similar values to the maximum value ($V_{b,s}$), making the options along the line relatively similar, which results in an overall low index. Hence, the higher is our index, the easier it is to make a choice, suggesting that trials with index values closer to 0, are the more difficult choices. Hence, we refer to this index as a choice simplicity index. The normalization with $\text{Endowment}_{b,s}$ of the chosen bundle is aimed to minimize the problem of over-scoring trials with higher endowments. In such cases, the budget line is longer (further away from the origin), and therefore $\Delta V_{i,b,s}$ is bigger for the mere fact that the distance between bundles is bigger.

**Controlling for changes in behavior**. We control for an over-estimation of trial-specific MMI values due to changes in behavior across blocks, which may increase index values, though in fact subjects simply changed their preferences between blocks and thus were actually consistent as long as their preferences were stable. For every subject, we computed four different aggregate MMIs, based on the 27 trials in each block, rather than the 108 trials of the entire experiment. We implemented the Leave-One-Out procedure on the aggregate MMI of each block to generate trial-specific MMI-blocks. Accordingly, we also recovered different utility functional parameters for every block and computed the trial-by-trial SV, with respect to the block-specific parameters (SV-blocks). We ran the same RFX-GLM, and used trial-specific MMI-blocks and SV-blocks as our inconsistency and value modulation regressors, respectively.

Moreover, we classified subjects' choice behavior in each block, and identified eight subjects (out of 33) who changed behavior across blocks (Supplementary Table 1). We thereafter ran the same RFX-GLM as in our main analysis (see Whole-brain analysis of choice inconsistency section), but this time used trial-specific MMI-blocks and SV-blocks only for the eight subjects who switched strategies. For the rest of our sample, we used trial-specific MMI and SV.

**Controlling for the misspecification**. In order to rule out the possibility that our results reflect the MMI's misspecification element, we repeated our main analysis,

using a different functional form. Halevy et al.[45] show that changing the functional form varies the misspecification element of the MMI, but the inconsistency element remains unchanged. Thus, we elicited subjects' preferences using a constant absolute risk aversion (CARA) utility index with an absolute risk aversion parameter $A$ (rather than CRRA utility index):

$$\omega(z) = 1 - e^{-Az}; A > 0 \qquad (9)$$

**Between-subjects analysis**. We conducted a standard between-subject analysis and examined if the brain areas that we found in our trial-by-trial analysis would show up also in a basic between-subject analysis. We ran an RFX-GLM with one predictor, a dummy for the trial identity. We ran the model in all our predefined ROIs, i.e., vmPFC, dACC, bilateral vStr and PCC (see Region of Interest Analysis section for details about the masking), as well as the mPFC/ACC cluster that was correlated with the trial-specific MMI (MMI ROI). For each subject, in each ROI, we extracted the average GLM-coefficient $\beta$ over the course of the entire experiment, to account for the average change in BOLD signal. We then correlated the average change in BOLD signal in each ROI with aggregate-level inconsistency indices—aggregate MMI, Afriat index, and number of GARP violations (see Aggregate inconsistency indices section for details). We corrected our analysis for multiple comparisons, using Bonferroni correction, and set $p_i \leq 0.05/(18$ comparisons) = 0.0028 as our statistical threshold (Supplementary Table 3).

**Functional localizers**. In the motor imprecision functional localizer, subjects were presented with linear graphs, and had to reach a black target using a trackball. It resembled the main task, but excluded any numerical or value representation (Supplementary Figure 1a). Subjects completed 27 trials, and had a maximum of 6 s on each trial to reach the black target, followed by a variable ITI of 6 s (jittered between trials).

Similarly, in the numerical execution imprecision localizer, subjects were presented with linear graphs, and had to reach a target {x,y} coordinates (Supplementary Figure 1b) on the graph. A title read their current {x,y} cursor position at the top of the screen. The numerical localizer resembled the main task, but excluded any value-based decision. Subjects completed 27 trials, and had a maximum of 12 s on each trial to reach the {x,y} coordinates, followed by a variable ITI of 9 s (jittered between trials). In both localizers, the graphs were varied and randomized across trials and subjects. We calculated motor and numerical imprecisions as the Euclidean distance between the cursor position at the moment the subject clicked the trackball, and the predefined target. In order to identify the neural correlates of the motor/numerical imprecision, we used the trial-by-trial motor/numerical imprecision as a predictor in an RFX GLM. Other predictors included a boxcar epoch function for the trial duration to model RT, as well as the graph's slope. All these predictors were entered for the trial duration, normalized and convolved with the HRF. In addition, six motion-correction parameters and the constant were included as regressors of no interest to account for motion-related artifacts.

## Data and code availability

The computer code used for the computation of MMI and the other inconsistency indices is available as an open source code at https://github.com/persitzd/RP-Toolkit. The datasets generated and/or analyzed during the current study, statistical maps and the rest of the computer code used to analyze further behavioral results, the imaging data and NRUM is available on OSF at https://osf.io/8jdfh/.

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

## Acknowledgements

We thank Y. David, A. Shuster and O. Ossmy for their helpful assistance. We also thank Y. Halevy and I. Saporta for fruitful discussions. This work was funded with grants from the Israel Science Foundation (#1104/13 for D.J.L. #1390/14 for D.P.), a joint grant to D.J.L. and D.P. from the Coller Foundation and the financial support of the Henry Crown Institute of Business Research in Israel.

## Author contributions

D.J.L. and D.P. conceived the study. D.J.L., D.P., and V.K designed the study. V.K. collected and analyzed the data. D.P. and R.W. performed analysis. D.J.L, D.P., R.W., and V.K. wrote the paper.

## Additional information

**Competing interests:** The authors declare no competing interests.

