## [Transparent Peer Review File · Nature Communications]

Reviewers' comments:

Reviewer #1 (Remarks to the Author):

Understanding why humans are sometimes inconsistent in their choices is a key issue within psychology and economics. To address this issue, Kurtz-David and colleagues propose that choice inconsistency is tied to neural noise creating variability in the valuation process. They evaluate this hypothesis by adapting a task from economics (developed by Choi) and developing a novel computational model that allows them to quantify trial-to-trial fluctuations in choice inconsistency and correlate that variability with brain responses recorded from fMRI. Interestingly, they find that activation within the VMPFC is correlated with both the severity of choice inconsistency and the subjective value of the chosen option.

In my view, the authors address an important question and put forth compelling results that stand up against several alternative explanations. Nevertheless, I believe the paper also suffers from a few shortcomings that could be addressed in a revision. These issues are discussed below. Given that I am not in a position to fully evaluate the computational modeling, my review will be limited to the fMRI analyses/interpretation and the implications of the results.

Major Comments

1) The authors do an excellent job of addressing potential alternative explanations, which is a key strength of the paper. However, I think there are at least two other factors that merit additional consideration:

a) First, please consider the role of confidence, which is intertwined with valuation and evokes similar responses within VMPFC (Lebreton et al., 2015, Nature Neuroscience). Given that trial-to-trial fluctuations in confidence are linked to VMPFC responses and decision-making processes, it seems likely that confidence could be tied to choice inconsistency. Considering this point may not require additional data/analyses, but some additional discussion could be useful for some readers.

b) Second, please consider the role of response times and choice difficulty (e.g., Shenhav et al., 2014, Nature Neuroscience). Recent work has suggested that brain systems tied to cognitive engagement explain framing biases better than emotion (Li et al., 2017, JNeurosci). This construct of cognitive engagement, whether considered through response times or choice difficulty, does not appear to be considered in the calculation of MMI. In addition, the manner in which RT is accounted for within the fMRI analysis is unclear, but it seems as though it assumes RT affects the height of the response rather than the duration, which some authors have argued to be suboptimal (e.g., Grinband et al., 2011, NeuroImage). Comparing the results with and without consideration of RT could also be informative (Yarkoni et al., 2009, PLoS One).

2) Part of the novelty of the paper hinges on the distinction between within-subject/trial-by-trial analyses and between-subject analyses. Although the trial-by-trial analyses are potentially more informative than a between-subject analysis, this point is not clear or convincing within the manuscript. The authors should consider developing this point more (perhaps in the Discussion) so that the results do not appear incremental. It is good to see the control analysis that addresses this point, but I think the authors should do more to help develop the reader's intuitions for how and why these two approaches might be similar or dissimilar in some contexts.

Minor Comments

1) The task is unclear. What are the subjects choosing and why? If the subjects had 12 seconds to

respond, what did they see after they make their response? Do they continue to look at the decision screen until 12 seconds has passed? Perhaps some additional explanation of the timing and structure of the task could be added to the main text and figure caption, space permitting.

2) The authors omit some relevant citations, including a seminal paper relating trial-to-trial variability in behavior to intrinsic connectivity (Fox et al., 2007, *Neuron*) and a recent paper that examines the interplay between reward and noise (Manohar et al, 2015, *Current Biology*). In addition, a recent brain stimulation paper lends credence to the idea that connectivity between regions may play a critical role in the precision of value-based choice (Polania et al., 2015, *Nature Communications*).

3) References to “a priori hypotheses” and “a priori ROIs” should be cut without reference to a pre-registration or other verification.

4) The VMPFC region in this paper seems more dorsal than related work in decision neuroscience (Figure 4), but I wonder if this perception is due to the fact that the authors present their data on single-subject brains? Please consider displaying results on a group average (e.g., MNI 152 brain).

5) The authors state they used multi-echo fMRI but they only mention one TE (30 ms) and they do not describe methods for combining information across multiple echo times.

6) Methods state that the fMRI analyses include 12 regressors but the figure captions do not agree with this number. In addition, it is unclear how RT was modeled (see above).

7) It is interesting to see that VMPFC responds to both subjective value and the degree of choice inconsistency. However, this is also a confusing finding because it is not clear if VMPFC activation is more strongly tied to one regressor or the other regressor. Did the authors perform any contrasts to tease apart this ostensible overlap? Alternatively, did the authors perform alternative neuroimaging analysis to assess connectivity? Recent meta-analytic work has shown that a given region (e.g., VMPFC) can be connected to distinct regions/networks depending on task context (Smith et al., 2016, *HBM*).

8) Although these results are interesting and potentially important in the field of cognitive neuroscience, it is unclear whether/how they might have implications for economics. This point could be strengthened within the Discussion section.

9) The authors state they will make their data and code available upon “reasonable request”, which may need some further qualification and clarification. Instead of making the data available upon request, I encourage the authors to make as much of their data/code available following publication. (Some reviewers might even reasonably request to evaluate code as part of the review.) At a minimum, please consider uploading thresholded and unthresholded statistical maps to NeuroVault to facilitate meta-analytic work and enable comparisons with related work.

Reviewer #2 (Remarks to the Author):

A wealth of evidence in the behavioral sciences suggests that human behavior is noisy, or inconsistent. There is also a growing understanding in neuroscience regarding how the brain makes everyday economic decisions. However, this paper makes an argument that little work exists to link these two literatures. In other words, what can our understanding of how the brain makes choices tell us about how the brain contributes to inconsistent/noisy choice behavior? To answer this question, the paper looks at a risky choice task conducted in an fMRI scanner. The paper presents a rigorous analysis of the behavioral data and constructs a novel metric of choice

inconsistency to use in the analysis of the fMRI data. The fMRI results point to the “value system” of the brain as a key driver of choice inconsistency.

The authors have constructed a paper that is quite satisfying to read. It brings together concepts from economics and neuroscience, and it presents the data in very nice figures. I am also fairly confident that researchers from a variety of academic backgrounds would find this paper interesting.

After reading the paper, I was left with several large questions. Very broadly, I am not sure that in fact any “mechanistic” evidence is presented in the paper. There is compelling “computational” evidence, but not sufficient evidence beyond the already well-established “value system” in the brain. Second, while I actually like the task very much and it is relatively novel compared to most fMRI studies, it may come at some cost because of its complexity. I outline more details below.

There are several missing citations, which I think would require changing some of the language regarding the paper being the “first” in a line of search.

Most prominently, it was strange to not see a discussion of Padoa-Schioppa (2013) in *Neuron*. That paper is not human data, but it is OFC data and focused exclusively on choice variability in the same choice sets.

Similarly, Polania et al. (2015) in *Nature Communications*. This is human data and very much is focused on understanding the neural mechanisms of choice accuracy/variability.

A recent theoretical review in *Nature Neuroscience* (Hayden and Hunt) also gets directly at underlying mechanisms for value-based choice and hence choice inconsistency.

Another paper involving fMRI and human data worth citing and possibly discussing is Grueschow et al (2015) in *Neuron*. In particular, that paper discusses posterior cingulate cortex, which is absent in the current paper.

While the NRUM model presents an account for how the choice inconsistency can arise, I was still left wondering if what is being shown is choice difficulty (e.g. as Shenhav et al. in 2014 showed) in ACC. Using the fitted utility functions it should be possible to set up an alternative GLM with this as a parametric regressor. Have the authors tried this? It would strengthen their claim(s) to be able to really rule out choice difficulty.

On a somewhat related note, I would like to see subject-level correlations between trial-to-trial SV and the trial-to-trial MMI index.

Figure 5ABC is helpful. I wonder if it should come earlier in the paper, so that the reader understands the implications, before seeing the fMRI results? I found Figure 5D a bit more confusing. I'd suggest a separate figure for $N > 2$.

As I mentioned earlier, I like the task. It differs from most neuroeconomics fMRI tasks. However, what exactly does it mean for the choice to be “difficult”? It is easy in a standard 2AFC task. It is not so obvious to me that it is the slope of the line, or rather that the slope can be interpreted as a continuous measure of choice difficulty. How do the authors think of choice difficulty here?

The authors used BrainVoyager for fMRI analysis. The MMI was listed as the first regressor. Does the order matter here? What if SV had been first? Basically, I am asking about orthogonality and/or any other partitioning of the variance that might be assumed in the model. Please list these assumptions, if there are any.

On page 11 the authors state “note that our Trial-specific-MMI index also grows as choices become more inconsistent.” This is then tied into the fMRI results in Figure 4. However, while this does

support the NRUM prediction in Figure 5, it is not direct evidence for such a variable. As it stands, the fMRI evidence shows correlates of SV and MMI in two brain regions, but there is no mechanistic distinction or any sort of “double dissociation” between regions or conditions. There are no individual fMRI differences shown. Given the claims of the paper, the authors should provide more fMRI evidence for their proposed model of choice inconsistency.

Additional comments

How many choices were “extreme” choices in the sense that corner choices were made?

What was in the questionnaire to verify subjects understood the task? Did everyone answer these questions correctly?

I appreciated the detailed plots presented in the supplement. However, some sort of figure or table summarizing the utility parameters recovered should be placed in the main paper. I think Choi et al have some scatter plots and tables. This is important, as the reader should be able to quickly know if most subjects were risk-averse, risk-neutral, etc.

The Figure 1 caption (lines 3,4 in the caption) is written in a confusing way. The parentheses around MMI_D and MMI_D-I could be interpreted as a function. I read it this way the first time. I suggest rewriting the sentence to avoid needing the parentheses.

Reviewer #3 (Remarks to the Author):

Kurtz-David and colleagues devised a new method for quantifying inconsistent economic choice on a trial-by-trial basis (their trial-specific MMI index). They find that BOLD activity in VMPFC and ACC was correlated with value, but also with the severity of inconsistency in choice. They interpret this overlap of activations as evidence suggesting that irrational decisions arise in the same brain areas responsible for value computation, possibly reflecting neuronal noise inherent to value computation.

The authors’ novel parametric index of the severity of inconsistency is clever, original and intriguing, and their results are exciting. The study should be of interest to a broad readership. However, I have several concerns regarding their data analysis and interpretation, some potentially grave, that need to be addressed by the authors before the manuscript can be considered fit for publication.

1) The core result of this study is that both subjective value and the severity of choice inconsistency are correlated with activations in VMPFC and ACC, as identified by parametric regression and conjunction analyses. This conclusion would be justified if value and inconsistency estimates predicted VMPFC/ACC activity independently. However, this is likely not the case. I suspect a certain degree of collinearity between the value and trial-specific MMI regressors because violations of GARP in trials with high-value alternatives (or large differences in value between both alternatives, respectively) will contribute more to the aggregate extent of inconsistency than GARP violations in low-value alternatives. Since the trial-specific MMI index is the difference between the aggregate MMI and the MMI in the partial dataset when applying a leave-one-out procedure, inconsistent choice should be associated with larger trial-specific MMIs in high value trials than in low value trials. In other words, value estimates and the severity of choice inconsistency indices should correlate. This raises the question if the authors’ conjunction analysis really revealed a true overlap of two independently identified activations, or whether the authors merely identified the shared variance between both regressors. The latter possibility would represent a strong challenge to their main conclusion.

The authors need to specify to what extent their value estimates and the trial-specific MMI

estimates are correlated. If they are correlated, they need to run additional analyses to address this collinearity issue. They could either reduce the collinearity between both regressors by orthogonalizing them (but they need to be careful when interpreting their results as orthogonalization usually assigns the shared variance to one regressor, and the residual variance to the other), or by identifying the unique variance associated with one regressor after partializing out the variance explained by the second regressor, and repeating this procedure with the second regressor after partializing out the variance explained by the first regressor.

2) The authors propose that inconsistent choice behavior is due to neuronal noise that induces variability in the computation of value. Even though I am generally sympathetic to this idea, the authors need to be careful to avoid overly strong claims not supported by their data, and they should consider other interpretations, too. The problem is that the authors measured variability in choice/BOLD, but not true neural noise per se. This means that they cannot rule out that trial-by-trial variability in value might actually not be due to neuronal noise, but arises from other processes, such as the use of cognitive heuristics, fluctuations in attention etc. that might also explain variability in value, and, by extension, the random term parameter e in their random utility model on p. 11. Some people go as far as stating that there is no such thing as neuronal noise. The authors need to state that, although their data are consistent with their noise model, they might be consistent with other accounts, too. Hence, they do not provide direct evidence for their model, but their model merely offers one of many explanations of the variability in value. I need to add that toning down their claims does not limit the originality of their data, it just calls for necessary caution in data interpretation.

3) The authors' value/MMI-related VMPFC cluster identified in their whole-brain GLM is quite dorsal and, in fact, more within medial than ventromedial PFC. A quick Neurosynth search revealed that this region is not typically associated with value, but with a number of other processes instead (default mode network, memory, executive functions etc.). How do the authors reconcile their medial PFC result with their value claims?

4) Discussion, p. 16: the abbreviation DDM should be defined on first use.

Concerns raised by Reviewer 1:*Major comments*

1. The authors do an excellent job of addressing potential alternative explanations, which is a key strength of the paper. However, I think there are at least two other factors that merit additional consideration:

- a) First, please consider the role of confidence, which is intertwined with valuation and evokes similar responses within VMPFC (Lebreton et al., 2015, Nature Neuroscience). Given that trial-to-trial fluctuations in confidence are linked to VMPFC responses and decision-making processes, it seems likely that confidence could be tied to choice inconsistency. Considering this point may not require additional data/analyses, but some additional discussion could be useful for some readers.

Response:

We thank the reviewer for the kind words, and think he/she has made an excellent point regarding the role of confidence in decision making. Indeed, low levels of confidence may lead to choice inconsistency. Lebreton et al. (2015) suggested that confidence "is quadratically related to first-order judgements". Following Lebreton et al., we modeled confidence as the second polynomial order of Subjective Value (SV). While trial-specific-MMI is correlated with SV^2 [$b=-0.000027$ $P<0.001$ in a clustered regression], the R^2 of the model is very low ($R^2 = 0.0395$), therefore confidence explains very little of the variation in choice inconsistency in our data. Similar to Lebreton et al., we verified the robustness of our result by adding SV^2 as an additional regressor in the RFX-GLM model, together with the first polynomial order of SV and trial-specific-MMI. We found that our main result for the neural correlates of choice inconsistency holds [RFX GLM, $n=33$, $p<0.0005$, cluster size correction, model regression: $BOLD = \beta_0 + \beta_1 trial + \beta_2 MMI_{trial_specific} + \beta_3 SV + \beta_4 priceratio + \beta_5 endowment + \beta_6 SV^2$. 6 additional motion-correction regressors were included as regressors of no interest]. The figure below depicts the neural correlates of the Trial-specific-MMI with the BOLD signal after controlling for confidence levels. We added a description of the new result under the "Ruling out alternative explanations" section. We also added the figure as Supplementary Fig. 10 to the Supplementary Materials.

- b) Second, please consider the role of response times and choice difficulty (e.g., Shenhav et al., 2014, Nature Neuroscience). Recent work has suggested that brain systems tied to cognitive engagement explain framing biases better than emotion (Li et al., 2017, JNeurosci). This construct of cognitive engagement, whether considered through response times or choice difficulty, does not appear to be considered in the calculation of MMI. In addition, the manner in which RT is accounted for within the fMRI analysis is unclear, but it seems as though it assumes RT affects the height of the response rather than the duration, which some authors have argued to be suboptimal (e.g., Grinband et al., 2011, NeuroImage). Comparing the results with and without consideration of RT could also be informative (Yarkoni et al., 2009, PLoS One).

In the original manuscript, we controlled for cognitive engagement by modelling RT and incorporating the price ratios (slopes) of the budget sets into the RFX-GLM model. We thank the reviewer for directing us to think more

carefully on the role of cognitive engagement in our results. We fully agree with what was shown in Li et al., (2017) regarding the notion that framing biases do not occur due to a competition between emotion and control but rather because of different cognitive engagements. Note, however, that in our task we show that inconsistencies arise even in cases where the framing is stable across trials. We separated our reply into two parts. In the first part we describe our treatment for the choice difficulty concern while in the second part we elaborate on our modeling of response time.

Choice difficulty:

Estimating the choice difficulty of each trial in our task is not straight forward. Importantly, the difficulty level in our task is an interaction between subject's own preferences and the characteristics of the specific decision problem (i.e. the slope of the budget set). For example, a risk-averse subject might struggle with steep slopes, as the temptation for allocating all tokens to one account increases. A risk-seeking subject, on the other hand, will have difficulties with moderate slopes since the two accounts provide similar opportunity. Therefore, a direct measure for the task difficulty must consider subject's elicited parameters.

In 2AFC tasks, choice difficulty is usually considered to be the difference in subjective values between the two options (ΔSV) – the smaller ΔSV is, the more similar the options are, and the difficulty level is higher (see Shenhav et al., (2014), Krajbich, Armel & Rangel, (2010), Basten et al., (2010), Krajbich et al., (2015), for several examples). Since each decision problem in our task is continuous, we cannot simply use ΔSV as a choice difficulty index. To address this issue, we propose an index for choice difficulty (similar in essence to the existing literature) which considers both subject's elicited parameters and the continuum of the budget set. For each subject, we discretized each budget line into 1,000 possible bundles. We then calculated the subjective value $v_{i,b,s}$ of each bundle i along each budget set b faced by subject s using the parameters elicited to this subject by the MMI elicitation method. Next, we subtracted $v_{i,b,s}$ from the maximal subjective value of that budget line. This yielded a $\Delta V_{i,b,s}$ measure for each bundle along the budget set (this step is analogous to taking the difference between two options in an 2AFC design). We then averaged all the $\Delta V_{i,b,s}$ across the 1,000 bundles, and normalized the result by the endowment $t_{b,s}$ of the given budget set to be able to compare across trials. Formally, denote $V_{b,s} = \max_{i \in [1, \dots, 1,000]} v_{i,b,s}$. Then:

$$\text{Choice simplicity}_{b,s} = \frac{\sum_{i=1}^N [V_{b,s} - v_{i,b,s}]}{N \cdot \text{Endowment}_{b,s}}$$

where $i \in [1, \dots, 1,000]$ is the bundle ($N=1000$); $b \in [1, \dots, 108]$ is the decision problems and $s \in [1, \dots, 33]$ is the subject.

Note that the higher our index is, the easier the choice is, suggesting that trials with index values closer to 0, are the most difficult choices. The intuition for this is that difficult choices occur when the bundles along the budget line (and hence the average of them) have similar values to the maximum value ($V_{b,s}$), making the options along the line relatively similar, which results in an overall low index. In contrast, when the value difference between the maximum bundle and all other bundles along the line is high (overall high index), the choice is easier. The normalization with $\text{endowment}_{b,s}$ prevents over-scoring of difficulty levels for trials with higher endowments. In such cases, the budget line is longer (further away from the origin), and therefore the difference is bigger for the mere fact that the distance between the bundles is bigger.

After calculating the choice simplicity index of each trial, we first considered the relationship between this index and reaction time. As expected, more difficult choices lead to longer RTs ($\beta = -2.782$, $p < 0.0001$, clustered regression by subjects). Moreover, we do not find evidence for a quadratic relationship between choice difficulty and RT - meaning, the RTs are not different for choices of intermediate difficulty ($p = 0.440$ for the second-polynomial order of the index when added to the model). For a subject level analysis, we divided the trials into 5 quantiles of choice difficulty, and plotted against RT (see Appendix A at the end of this letter). No distinct pattern emerges from this individual level analysis – the relationship between RT and choice difficulty are linear for some subjects (e.g. subjects 404 and 204) and absent for others (e.g. subjects 203 or 410).

Next, we examined the relationship between the choice simplicity index and the inconsistency index (Trial-specific-MMI). Trial-specific-MMI is negatively correlated with the simplicity index ($\beta = -0.0302$, $p < 0.0001$). Meaning that the more difficult is the choice problem, the higher is the corresponding inconsistency level. However, the simplicity index also explains little of the variance in Trial-specific-MMI scores ($R^2 = 0.0137$). It is important to note that the random utility model that we propose predicts these findings. If there was no noise, then even the most difficult choice would not lead to inconsistencies (i.e. choice difficulty alone cannot lead to choice inconsistency). It is exactly the noise that "connects" choice difficulty to inconsistency and an obvious implication is that more difficult problems are more likely to produce inconsistency (for the same amount of

noise), meaning that a (weak) negative correlation between the choice simplicity index and the level of inconsistency is expected.

In any case, to formally rule out the possibility that our main findings are mainly influenced by choice difficulty, we included the choice simplicity index in our general RFX-GLM, and were able to replicate our main result for the trial-specific-MMI and SV clusters. The figure below presents a replication of Figure 4a-d of the manuscript, this time controlling for choice difficulty in the regression [$n=33$, $p<0.001$, cluster size correction, $x=0$, model regression:

$BOLD = \beta_0 + \beta_1 trial + \beta_2 MMI_{trial_specific} + \beta_3 SV + \beta_4 priceratio + \beta_5 endowment + \beta_6 choice_simplicity$. 6 additional motion-correction regressors were included as regressors of no interest].

Moreover, we examined the RFX-GLM when excluding the SV and endowment regressors, due to the expected high collinearity of the two predictors with the choice simplicity index ($R\text{-square} = 0.5225$, clustered regression). When we exclude these two regressors, we were able to replicate our main result for the neural correlates of Trial-specific-MMI in the same threshold reported in the manuscript ($p<0.0005$, cluster-size corrected). Importantly, we found that our index for choice difficulty is correlated with ACC activation (and a few other brain areas), as previously reported in the literature (e.g. Shenhav et al., 2014 and Paus et al., 1998). The figure below reports these findings [$n=33$, $p<0.0005$, cluster size correction, model regression: $BOLD = \beta_0 + \beta_1 trial + \beta_2 MMI_{trial_specific} + \beta_3 priceratio + \beta_4 choicesimplicity$. 6 additional motion-correction regressors were included as regressors of no interest].

We have also repeated the ROI analysis, and similarly to our original results, we found that Trial-specific-MMI is positively correlated with dACC, vmPFC, and PCC activations ($q(FDR)<0.05$), even when controlling for difficulty level (see Figure below. For illustration and comparison purposes we set the threshold as in the whole-brain analysis. The regression model is the same as the whole-brain analysis). As in our original results, the SV regressor was also correlated with dACC, vmPFC and PCC activations.

To sum up, we introduced an index for choice difficulty in a setting of continuous budget lines and we used it to control for choice difficulty. Our main findings hold and are consistent with our interpretation that the trial-specific-MMI index captures the effect of trial-level variation in neural value signals: for two trials with the same difficulty level, a larger variation in the BOLD signal is correlated with higher inconsistency (the trial-specific-MMI). We have included the results of this analysis and the above discussion in the “Ruling out alternative explanations” section, added the analyses description to the Method section and added the relevant figures to the Supplementary Materials (Supplementary Fig. 9).

Modeling of RT:

We thank the reviewer for pointing out this issue, as it was not properly reported in the original manuscript. Following Grinband et al. (NeuroImage, 2008), we modeled RT for the trial duration - we used Grinband et al. variable epoch approach, which models each trial with a boxcar epoch function, whose duration is equal to the RT of the trial. A single regressor was then constructed from these boxcars to use in the GLM, and convolved with a canonical HRF. We now clarify this issue in the Methods section of the manuscript as well. However, following Reviewer 1’s suggestion, we have also run the RFX-GLM model without the RT regressor. Our main findings do not hold when we neglect the modeling of RT. This is due to the fact that the RT predictor captures all RT-related activations, which reflect the trial occurrence. Such activations include all the visual, motor and cerebellar activations, as pointed out by Yarkoni et al. (2009).

2. Part of the novelty of the paper hinges on the distinction between within-subject/trial-by-trial analyses and between-subject analyses. Although the trial-by-trial analyses are potentially more informative than a between-subject analysis, this point is not clear or convincing within the manuscript. The authors should consider developing this point more (perhaps in the Discussion) so that the results do not appear incremental. It is good to see the control analysis that addresses this point, but I think the authors should do more to help develop the reader’s intuitions for how and why these two approaches might be similar or dissimilar in some contexts.

Response:

We appreciate the reviewer’s comment and agree that we should have given more attention to this issue and its merit to our study. The main point here is that without the trial-by-trial variability, one cannot capture the noisy computational process. When using only a between subject analysis, variations in the BOLD signal across trials are averaged, and deviations or irregular patterns in subject’s responses are missed. Such choices are actually the most informative data points, as they represent the biggest deviations from rational choice.

Because we feel this contribution is so important, we have expanded the discussion in multiple places in the manuscript. We now address the limits of a between-subjects analysis in the introduction, and have added to the Results a section titled “Motivation for using Trial-specific-MMI” where we report the results of a standard between-subject analysis using the aggregate indices. The null results demonstrate that using our novel trial-specific inconsistency index improves statistical power in a systematic search for the neural correlates of inconsistency because it allows us to analyze the irregular neural patterns in subject’s response on an inconsistent choice trial. We clarified the contribution in the discussion, and note that our index can be used in standard behavioral experiments as well. We hope that now the manuscript better emphasizes the importance and advantages of our trial-by-trial approach compared to a between-subject analysis.

Minor Comments:

1. The task is unclear. What are the subjects choosing and why? If the subjects had 12 seconds to respond, what did they see after they make their response? Do they continue to look at the decision screen until 12 seconds has passed? Perhaps some additional explanation of the timing and structure of the task could be added to the main text and figure caption, space permitting.

Response:

We apologize if the description of the task was unclear. We have now revised Fig. 2 and the Methods to simplify and clarify the task presentation.

Each decision problem presented the subjects with a different budget line. Each (x, y) coordinate along the budget line is a lottery where in 50% chance the subject could win X tokens and in 50% chance the subject could win Y tokens (1 token = 5 NIS, 3.5 NIS \sim 1 USD). The subject was asked to choose one of these lotteries from the line. At the end of the experiment, one of the trials was implemented (by randomly selecting either account X or Y). The subject executed this lottery by tossing a fair coin.

In the figure below, for example, if the subject chose Coordinate A (10,35) out of all possible lotteries, then if X was selected, she would win 10 tokens while if Y was selected, she would win 35 tokens. Another example is Coordinate B, where the subject wins 45 tokens if x is selected, and 10 tokens if y is selected. If the subject is extremely risk-seeking, she might choose the lottery where she can achieve the maximal possible payoff, which is the intersection of the budget line with the X -axis (Coordinate C, as X is the cheaper account). This subject has a 50% chance of winning 60 tokens, but she might win 0 tokens, if Y is selected. By contrast, a risk-averse subject, will choose the coordinate which intersects with the 45 degrees line from the origin (Coordinate D). This bundle is a lottery with equal payoffs, as it comprises of the same amount of tokens for both X and Y . These are four examples of the bundles subjects can choose along the budget line. The slopes of the budget lines and endowments (distance from the origin) varied across trials.

Each decision problem was presented for a maximum of 12 seconds, and subjects could make their choices freely within this 12 seconds time-window. If the subject made her choice before the end of the 12 seconds time-window, the screen changed to the ITI screen (a fixation cross in the middle of the screen) and the remaining time was added to the time of the ITI (which was 9 seconds). This was repeated for 108 trials (divided into 4 equally long blocks).

We thank the reviewer for the comment, and now we clarify these issues in the main text, in the caption of Figure 2, and in the Methods section. The figure below has been added as a new panel in Figure 2, and we included in the Supplementary Materials an English translation of the experimental instructions and the pre-scan questionnaire subjects answered outside the magnet. We hope that this clarifies the task.

2. The authors omit some relevant citations, including a seminal paper relating trial-to-trial variability in behavior to intrinsic connectivity (Fox et al., 2007, Neuron) and a recent paper that examines the interplay between reward and noise (Manohar et al, 2015, Current Biology). In addition, a recent brain stimulation paper lends credence to the idea that connectivity between regions may play a critical role in the precision of value-based choice (Polania et al., 2015, Nature Communications).

Response:

We apologize for this, and have incorporated all these insightful works into the manuscript.

We now discuss Fox et al. (2007) in the Discussion and state that they show that intrinsic resting state activity accounts for most of the variability in behavior. They conclude by stating that inconsistency in performance can be due to fluctuations in intrinsic neuronal behavior. We believe that our findings regarding the correlation between neural variability and inconsistency in choices demonstrates their prediction.

We now cite Manohar et al. (2015) in our very last paragraph of the Discussion. They show that noise can be attenuated by increasing rewards and we interpret their result as an evidence for a tradeoff between the metabolic costs for decreasing neural variability and gaining higher speed and precision in decision making.

We now discuss Polania et al. (2015) in the Discussion, and suggest that their findings are compatible with our proposed mechanism for choice behavior. We claim that in the absence of value-related regions (as in lesioned patients) or the disruption of the synchronization between them (using brain stimulation), choices are highly inconsistent (as Polania et al. (2015) report) because of the limited ability of the brain to compute the valuations necessary for consistent choice behavior. When these regions are intact, value signals can be computed, but with a degree of variability inherent to neural computation.

3. References to “a priori hypotheses” and “a priori ROIs” should be cut without reference to a pre-registration or other verification.

Response:

We completely agree. We thus refined the text, and stated we chose these ROIs based on previous studies, without mentioning any “a priori” hypotheses.

4. The VMPFC region in this paper seems more dorsal than related work in decision neuroscience (Figure 4), but I wonder if this perception is due to the fact that the authors present their data on single-subject brains? Please consider displaying results on a group average (e.g., MNI 152 brain).

Response:

Following the reviewer’s suggestion, we now present all the brain figures using the MNI-152 brain template. We agree that the 152-MNI brain template is more suitable for presenting group average results. In the whole-brain analysis, our results are indeed more dorsal than usually reported, and hence we changed the text to state mPFC accordingly.

However, importantly, in our ROI analysis, we find a significant activation in the dorsal part of the vmPFC ROI that we defined based on the ROI reported in Bartra et al. (2013), which used more than 200 fMRI papers to define the value related activation in the vmPFC. Therefore, we think that in the ROI analyses it is suitable to refer to the activated region as part of the vmPFC.

5. The authors state they used multi-echo fMRI but they only mention one TE (30 ms) and they do not describe methods for combining information across multiple echo times.

Response:

We used a Multi-Band (and not multi-echo) EPI sequence (TR = 1.5s; TE = 30 ms; flip angle = 70°; matrix = 86 × 86; field of view (FOV) = 215 mm; slice thickness = 2.5 mm; band factor = 2). So there was only one echo time. We do apologize for the slackness, and thank the reviewer for his/her cautious and meticulous reading. We made the necessary changes in the methods and main text.

6. Methods state that the fMRI analyses include 12 regressors but the figure captions do not agree with this number. In addition, it is unclear how RT was modeled (see above).

Response:

We changed the text and figure captions to clearly state how many regressors we have in each analysis and describe each of them. Regarding RT, see the discussion regarding the modeling of RT above in our reply to your major comment 1b.

7. It is interesting to see that VMPFC responds to both subjective value and the degree of choice inconsistency. However, this is also a confusing finding because it is not clear if VMPFC activation is more strongly tied to one regressor or the other regressor. Did the authors perform any contrasts to tease apart this ostensible overlap? Alternatively, did the others perform alternative neuroimaging analysis to assess connectivity? Recent meta-analytic work has shown that a given region (e.g., VMPFC) can be connected to distinct regions/networks depending on task context (Smith et al., 2016, HBM).

Response:

We thank the referees for pointing out the need for clarifying our results. We wish to address this concern from multiple directions so that no doubt will remain regarding our main findings. We added to the manuscript a

new section in the Results titled “Dissociation if the SV and Trial-specific MMI regressors, with the discussion below.

First, we wish to clarify the form of the relationship between the subjective value of the chosen bundle and our Trial-Specific MMI index. According to the theory upon which the index is based, there is a relation between these regressors. Given a budget line, a chosen bundle, and a parametric utility function, a contradiction between the utility function and the revealed behavior arises if the chosen bundle does not maximize the utility function over the budget set. The MMI index measures the extent of this contradiction in a two-step procedure. First, the budget line is adjusted inwards until the contradiction vanishes. Second, we calculate the reduction in expenditure between the original budget line and the adjusted budget line in percentages (e.g. if the chosen bundle is the optimal bundle, the MMI index of this observation is zero). Intuitively, **the lower the value of the chosen bundle**, the lower is the adjusted budget line and **the higher is the reduction in expenditure** (bigger distance in percentages) between the original budget line and the adjusted budget line.

However, this negative correlation should have little bearing on our analysis - for a given bundle, the SV depends on the utility function and does not depend on the specifics of the budget line (prices and expenditure). On the other hand, the trial-specific-MMI depends on both. Therefore, knowing the SV on a given trial is insufficient for calculating the trial-specific-MMI. For this reason, we expect the SV and the trial-specific-MMI regressors to exhibit weak negative correlation.

Indeed, the Trial-specific-MMI and SV regressors are negatively correlated ($\beta = -0.00133$, $p < 0.001$ in a clustered regression by subjects). This correlation remains unchanged if we control for the expenditure and slope. Also, as expected, the R^2 of the regression is very low ($R^2 = 0.0496$), which means that less than 5% of the variance of the Trial-specific-MMI is explained by the SV regressor. Therefore, there is little collinearity between the two regressors (SV and MMI) which means the standard errors are not inflated. The significant activations in the same brain area is therefore evidence that each regressor represents a different but complementary feature of the decision-making process. We also refer the reviewer to the subject-level correlations presented in Appendix B below.

To demonstrate that the two regressors can be disentangled, we have conducted an orthogonality analysis, a contrast analysis, and a functional connectivity analysis.

In the orthogonality analysis we implemented the main GLM, but replaced Trial-specific-MMI with the residuals ($\tilde{\epsilon}$) from a regression of SV on Trial-specific-MMI (and vis-à-vis, run the main GLM, and replaced SV with \tilde{u}). Importantly, even after controlling for the weak collinearity, our main results still hold (see figure below), [$n=33$, RFX, $p < 0.0005$, cluster-size correction, model regressions:

(1) $BOLD = \beta_0 + \beta_1 \text{trial} + \beta_2 \tilde{\epsilon} + \beta_3 SV + \beta_4 \text{priceratio} + \beta_5 \text{endowment}$;

(2) $BOLD = \beta_0 + \beta_1 \text{trial} + \beta_2 \text{MMI}_{\text{trial_specific}} + \beta_3 \tilde{u} + \beta_4 \text{priceratio} + \beta_5 \text{endowment}$.

6 additional motion-correction regressors were included as regressors of no interest].

For the contrast analysis, we defined the two-sided (and more stringent) contrast $SV \neq \text{MMI}$, using the RFX-GLM reported in Figure 4a-d [$n=33$, $p < 0.0005$, cluster size correction, model regression: $BOLD = \beta_0 + \beta_1 \text{trial} + \beta_2 \text{MMI}_{\text{trial_specific}} + \beta_3 SV + \beta_4 \text{priceratio} + \beta_5 \text{endowment}$. 6 additional motion-correction regressors were included as regressors of no interest]. As can be seen in the figure below, there is a dis-junct activity of the two regressors in the mPFC/ACC region in the same region that we report in the general GLM. This demonstrates that there are voxels in these areas that encode SV information and MMI information but in a different manner.

Furthermore, as suggested by the reviewer, we also examined if the functional connectivity across brain areas is different between SV and MMI. We performed a psychophysical interaction (PPI) analysis to estimate the functional connectivity of the BOLD signal between our predefined ROIs (vmPFC, dACC and PCC), that were used as seed regions, with other brain regions, interacting with the SV and MMI regressors. Following Smith et al. (2016), and our conjecture that these two regressor represent distinct processes, we anticipated that given the different task context (i.e., the parametric Trial-specific-MMI vs. SV regressors) each seed region will reveal a different/distinct functional connectivity network. Hence, we did not define any target regions intentionally, as we wanted to obtain the full connectivity network for each task context.

The time series of the BOLD signal in each ROI was z-scored to generate the time series of the neuronal signal for each source region as the physiological variable in the PPI. We tested (separately) each parametric regressor, SV and Trial-specific-MMI, as the psychological variable. The psychological regressors were normalized and convolved with the canonical HRF and entered into the regression model. An additional regressor represented the interaction between the psychological and physiological factors, and showed areas with significant differential connectivity to each ROI. We modeled RT similarly to the whole-brain analysis, and used the price ratio of the budget set and the endowment as control predictors. These predictors were entered for the trial duration, normalized and convolved with the HRF as well. In addition, six motion-correction parameters and the constant were included as regressors of no interest to account for motion-related artifacts. We used random fixed effects for the group-level analysis, and set the threshold to $p < 0.0005$ with cluster-size correction, similarly to the GLM analysis. We hence run 6 models in total (3 ROIs * 2 psychological contexts).

The results from the PPI analysis (see figure below) indicate that the seed regions have a functional connection to the motor and visual regions, which are probably related to task-execution. In addition, we found a connection to the value-related regions, such as the Insula and dIPFC (see Rangel, Camerer and Montague, 2008, and Levy and Glimcher 2012, for a review of these regions' involvement in value-based choice). Functional connectivity with other value-related regions does not seem surprising, as some scholars suggested that the process of value computation is a spread and distributed process, with many regions taking part (see Hunt & Hayden, 2017 for a review).

Under the SV context we found interactions within seed regions – specifically, the PCC and vmPFC interact with the dACC. Under the Trial-specific-MMI context, there were no interactions between the seed regions and the dACC. Such a pattern might indicate, that choice inconsistency is a spontaneous or random process, one that is not coordinated across the different nodes of the “value network”. In any case, we found clear separation between the functional connectivity networks of SV and Trial-specific-MMI, strengthening our observation that they capture different neural processes and cognitive measures.

Finally, we have extracted for each subject, in each ROI - dACC, vmPFC and PCC - the mean effect (beta value) for each predictor. We compared beta values using Wilcoxon sign-rank test. Our results are not significant for the vmPFC and dACC ROIs ($p=0.0504$ and $p=0.2877$, respectively), and significant for the PCC ROI ($p<0.005$). This suggests that we cannot reject the hypothesis that both predictors have, on average, an important and similar in magnitude effect on the BOLD signal in the vmPFC and dACC. The figure below shows the results of the analysis. We added the result of this analysis as Figure 4d.

We believe that together these analyses confirm that our main findings indeed reflect a true overlap of two regressors representing different computations, rather than their shared variance. We have added all these important analyses and results to the manuscript, and included the rest of the results in the Supplementary Materials.

8. Although these results are interesting and potentially important in the field of cognitive neuroscience, it is unclear whether/how they might have implications for economics. This point could be strengthened within the Discussion section.

Response:

We thank the reviewer for the opportunity to extend the discussion into the implications of this study to Economics. We added the main issues raised here to the discussion.

Economists are well aware that decision makers exhibit inconsistent behavior. A large literature (and at least two Nobel Memorial Prizes in Economic Sciences) is devoted to various cognitive scenarios in which decision maker typically behave inconsistently. Typically, economists provide several alternative explanations for such inconsistent behavior. Some attribute it to the bounded mental capacity or limited attention of decision makers and maintain that using “short-cuts” may lead to the inconsistencies when choice problems become more complex (like, for example, choosing a health or pension plan). Others attribute the inconsistency to misunderstandings of the choice problem (like the array of prices faced by a consumer at a grocery store). However, both of these explanations are difficult to reconcile in relatively simplified experimental designs which do not frame or manipulate choices in any particular way.

We think that the results described in this study support a different explanation for inconsistent choice behavior that relies a long tradition of Random Utility Models (RUMs) (for which McFadden won the Nobel prize in 2000). In Economics, a random utility model underlies almost all econometric models applied to behavioral data (e.g. for parametric estimation of demand curves). However, most of the literature remains agnostic regarding the nature and source of the noise in utility and simply attributes it to limitations in the data available to the econometrician (Manski 1977). However, more recently, economic theorists have proposed that the valuations which underlie choice may themselves be stochastic (e.g. Gul and Pesendorfer, 2006; McFadden 2005). While such theories do place constraints on the pattern of behavior we might observe, they are weak tests of the theory (by design) because utilities are assumed to not be observable. Filling this gap, Webb et al. (2018) proposed a source for the stochastic process of choice – the neuronal value computation that takes place in the Decision Maker’s brain. They suggest the Neuronal Random Utility Model, where during the value computation process, noise (arising from thermodynamic noise at the cellular and synaptic level) is attached to the true value, such that when it is time to compare values and make a choice the values include a stochastic component. The experiment conducted in the current study, provided a series of findings that strengthens this idea.

An additional contribution of this study to Economics is the implementation of the Money Metric Index at a trial-by-trial analysis. Given a dataset of choices from linear budget sets, various non-parametric indices,

based on revealed preference theory, were proposed in order to measure the extent of inconsistency. However, when researchers wished to estimate the preference of the decision maker, they used standard econometric methods based on the minimization of the distance between the observed and predicted bundles. Halevy et al. (2018) suggested the Money Metric Estimation procedure as a parametric estimation procedure based on the revealed preference principle. Moreover, they showed that the value of the objective function is the sum of a well-known inconsistency index (Varian Inconsistency index, see supplementary material for details) and a measure of the misspecification inherent in any parametric estimation. However, as all other indices based on revealed preference theory, the primitive was the dataset as whole and not each observation independently.

As we established earlier (major comment #2) for the current study it was imperative to come up with a trial-by-trial inconsistency score. The leave-one-out procedure provided us with the solution – produce a trial level score while still using the dataset as a primitive. Now, this methodology can be taken into standard behavioral laboratory experiments in Economics to test other theories that imply heterogeneous effects across trials (e.g. choice dynamics).

9. The authors state they will make their data and code available upon “reasonable request”, which may need some further qualification and clarification. Instead of making the data available upon request, I encourage the authors to make as much of their data/code available following publication. (Some reviewers might even reasonably request to evaluate code as part of the review.) At a minimum, please consider uploading thresholded and unthresholded statistical maps to NeuroVault to facilitate meta-analytic work and enable comparisons with related work.

Response:

We fully agree, and are sorry for this unfortunate sentence. That was not our intention. All statistical maps will be uploaded to NeuroVault, and all the code used for the analyses will be uploaded to GitHub. In fact, the code package used to calculate the inconsistency indices discussed in the paper, is already on GitHub (<https://github.com/persitzd/RP-Toolkit>).

Concerns raised by Reviewer 2:

Major comments

1. **Very broadly, I am not sure that in fact any “mechanistic” evidence is presented in the paper. There is compelling “computational” evidence, but not sufficient evidence beyond the already well-established “value system” in the brain.**

AND

On page 11 the authors state “note that our Trial-specific-MMI index also grows as choices become more inconsistent.” This is then tied into the fMRI results in Figure 4. However, while this does support the NRUM prediction in Figure 5, it is not direct evidence for such a variable. As it stands, the fMRI evidence shows correlates of SV and MMI in two brain regions, but there is no mechanistic distinction or any sort of “double dissociation” between regions or conditions. There are no individual fMRI differences shown. Given the claims of the paper, the authors should provide more fMRI evidence for their proposed model of choice inconsistency.

AND

On a somewhat related note, I would like to see subject-level correlations between trial-to-trial SV and the trial-to-trial MMI index.

Response:

Thank you for these comments. We acknowledge that we did not provide enough evidence to connect our empirical findings with the NRUM. First, we have changed the wording throughout the entire manuscript (including the title) to state that we are providing a computational model rather than a mechanistic account of how neuronal variability relates to inconsistent choice behavior. This was clumsily stated in the original text. Second, we took several steps to provide additional supporting evidence for the linkage between the trial-specific-MMI and our computational model. To briefly summarize, we present individual level analysis findings, present results on a dissociation between the SV and Trial-By-Trial MMI regressors (see our response to Reviewer 1 minor, point 7) and we demonstrate the relation between the computational NRUM model and our empirical findings.

Subject-level fMRI

We have run individual GLMs for each subject in each ROI (dACC, vmPFC and PCC) and looked for conjunct activations of Trial-specific-MMI with SV, to identify overlaps between value computations and deviations from rational choice on the individual level. For each subject we ran the same model ($BOLD = \beta_0 + \beta_1 \text{trial} + \beta_2 \text{MMI}_{\text{trial_specific}} + \beta_3 \text{SV} + \beta_4 \text{priceratio} + \beta_5 \text{endowment}$), though due to lack of statistical power we used FFX, and allowed for a liberal threshold of the conjoint probability, $p < 0.15$ (note that in a conjunction analysis we look for the minimum t -statistic across multiple comparisons, and hence the individual activation might have a much lower p -value (Nichols et al., NeuroImage, 2005)). We found conjunct clusters for 24 subjects from our sample in at least one of the ROIs. If we loosen the threshold to $p < 0.2$, we find clusters for 3 additional subjects. These results support the claim that the group-level overlap reflects overlap at the single-subject level. We hereby present the individual conjunct clusters in each ROI. We added a description of these results along with the figure and the detailed table below to the manuscript (Fig. 4f and Supplementary Table 5, respectively).

SID	vmPFC		peak voxel (MNI)			dACC		peak voxel (MNI)			PCC		peak voxel (MNI)		
	Overlap	Cluster size	x	y	z	Overlap	Cluster size	x	y	z	Overlap	Cluster size	x	y	z
103	V	111	8	52	-3	V	135	2	35	21	V	62	-18	-61	40
104	V	5454	10	42	0	V	243	1	38	19	V	492	-12	-48	40
202	-	-	-	-	-	-	-	-	-	-	V	38	0	-61	17
203	V	216	-10	46	-1	V	54	0	37	19	V	69	4	-62	28
204	V	54	-3	42	4	V	1593	1	38	26	V	615	-2	-36	33
205	-	-	-	-	-	-	-	-	-	-	-	-	-	-	-
206	V	5859	10	40	0	V	2997	2	36	23	V	5701	-9	-59	10
401	V	27	10	41	0	V	108	-5	32	22	V	120	-3	-46	13
402	-	-	-	-	-	-	-	-	-	-	-	-	-	-	-
403	-	-	-	-	-	V(p<0.2)	27	-9	44	25	V	991	6	-55	10
404	V	3996	10	45	0	V	675	2	34	22	V	3622	12	-42	31
405	V	5508	8	42	2	V	324	3	34	22	V	3565	-12	-37	28
406	-	-	-	-	-	-	-	-	-	-	-	-	-	-	-
407	V	162	-6	49	-17	V	27	-3	38	16	V	841	15	-46	4
408	V	27	-6	47	4	V	27	0	44	25	V	132	3	-55	19
409	V(p<0.2)	54	-4	47	-8	-	-	-	-	-	V	17	-12	-55	10
410	-	-	-	-	-	-	-	-	-	-	-	-	-	-	-
412	-	-	-	-	-	-	-	-	-	-	-	-	-	-	-
413	V	27	3	38	-8	-	-	-	-	-	-	-	-	-	-
414	V	675	10	55	-7	-	-	-	-	-	V	88	-7	-61	19
415	V(p<0.2)	27	3	35	-11	-	-	-	-	-	-	-	-	-	-
416	V	27	3	39	-18	V	135	-10	35	33	-	-	-	-	-
417	V	378	-1	39	-19	V	27	-3	29	22	V	329	-6	-58	40
418	V	54	-7	41	1	-	-	-	-	-	V	322	0	-55	6
419	-	-	-	-	-	-	-	-	-	-	V	2877	6	-61	10
420	V	810	1	48	3	V	837	2	41	26	V	1799	-3	-52	31
421	V(p<0.2)	54	9	43	1	-	-	-	-	-	-	-	-	-	-
422	-	-	-	-	-	V(p<0.2)	27	-9	30	29	-	-	-	-	-
424	V	918	6	52	-9	V	27	-6	44	31	V	279	3	-37	28
426	V	189	3	46	4	V	625	-9	38	26	V	726	3	-33	34
427	-	-	-	-	-	-	-	-	-	-	V	238	-6	-46	22
428	-	-	-	-	-	-	-	-	-	-	-	-	-	-	-
430	-	-	-	-	-	V(p<0.2)	27	3	36	22	-	-	-	-	-
Total	18 (21)					15 (18)					21				

Relating the NRUM to Trial-specific-MMI

To demonstrate that the NRUM is closely related to the observed neural activity, we simulated the valuation process of our subjects based on their observed behavior and checked the correlation between their random utility valuations (equivalent to the BOLD signal) and the inconsistency of their simulated choices (equivalent to the Trial-specific-MMI index).

We generated simulated datasets for each subject. To do that, in each trial, we reconstructed the set of bundles she encountered and calculated the subjective value using the parameters elicited by the MMI method. These correspond to the v_i s in the proposed model. We calibrated two skewed distributions (the zero mode Gumbel distribution and the zero-mean log normal distribution) for the neural noise e_i using the observed inconsistency level of each subject (the standard deviation of the distributions was chosen so that the average level of the Afriat Inconsistency Index will match the observed index). Based on the v_i s and the calibrated distributions for e_i we calculated the random utility values (\tilde{v}_i) for each alternative in every trial. For each trial (following value maximization), the chosen bundle was the alternative with the highest random utility value. We repeated this procedure, for each subject, 1000 times in each noise distribution. Hence, we obtained 1000 simulated datasets for each subject (in each of the two noise distributions).

Next, we tested whether the simulated data sets are compatible with our interpretation of the neural results. For each simulated trial, we calculated the noise of the chosen bundle as a proxy for the valuation noise in the BOLD signal and the Trial-specific-Afriat index as a proxy for the Trial-specific-MMI index (which we could not re-calculate after using the MMI elicited parameters to calculate the v_i s). We pooled these two series across simulations and trials and correlated them.

Importantly, for all our subjects we found a **significant positive correlation**, implying that the NRUM is consistent with our main empirical findings. Note that this result was established by using the degenerate, low variability, Trial-specific-Afriat index, meaning that if we knew the actual preferences of the subjects (rather

SID	Gumbel dist.	log normal dist.	SID	Gumbel dist.	log normal dist.
103	0.1245*	0.1484*	412	0.1084*	0.1105*
104	0.1345*	0.1568*	413	0.1201*	0.1089*
202	0.1048*	0.1341*	414	0.1273*	0.1662*
203	0.1233*	0.1261*	415	0.1227*	0.1175*
204	0.1290*	0.1581*	416	0.0543*	0.0455*
205	0.1266*	0.1288*	417	0.1084*	0.1391*
206	0.1345*	0.1369*	418	0.1312*	0.1500*
401	0.1244*	0.1343*	419	0.0934*	0.0916*
402	0.1002*	0.0914*	420	0.1120*	0.1339*
403	0.0963*	0.1139*	421	0.1303*	0.1301*
404	0.0607*	0.0706*	422	0.0779*	0.0701*
405	0.1148*	0.1143*	424	0.1053*	0.0889*
406	0.1196*	0.1187*	426	0.1240*	0.1259*
407	0.1289*	0.1620*	427	0.1200*	0.1133*
408	0.1208*	0.1298*	428	0.1281*	0.1746*
409	0.1227*	0.1484*	430	0.1365*	0.1563*
410	0.1206*	0.1207*			

(*) $p < 10^{-10}$

than eliciting them using the MMI) we could have obtained an even stronger result because of the higher variability of the trial-specific-MMI index. The table below provides the correlation per subject in both batches of simulations. We also added (in Appendix C at the end of this letter), for each subject, the distribution of the simulation-level correlations for the Gumbel distribution (the distributions for the log-normal are almost identical). We added a description of this simulation to the Methods section and added the results of the simulation (including the table below) to the main text and additional information to the Supplementary Materials.

2. Second, while I actually like the task very much and it is relatively novel compared to most fMRI studies, it may come at some cost because of its complexity. I outline more details below.

And

How many choices were “extreme” choices in the sense that corner choices were made?

And

What was in the questionnaire to verify subjects understood the task? Did everyone answer these questions correctly?

Response:

We thank the reviewer for his/her kind words. We agree that the task is not as simple as most fMRI tasks. However, we have taken several steps to verify subjects understood the task, and we have now simplified its description in the manuscript (Fig 2).

Before entering the fMRI room, we gave subjects a thorough instruction sheet to read. Our instructions included many examples and were written in simple terms to avoid confusion. In addition, we kept the instructions very similar to those used in many laboratory experiments in the last decade (e.g. Choi et al. (2007) and Halevy et al. (2018)). In addition to the instructions, we gave subjects a pre-scan questionnaire. The pre-scan questionnaire presented subjects with various decision problems. We asked them to identify the intersections with the axes; identify which account can provide more tokens (i.e. identify the “cheaper” lottery); pick a random coordinate on the budget line, explain what are the possible payoffs for this coordinate; and calculate monetary payoffs for different lottery results. Subjects completed the pre-scan questionnaire after reading the experimental instructions. After the pre-scan questionnaire, the experimenter went over their answers. In case the subject made a mistake, the experimenter explained the instructions orally, and then repeated the question in the questionnaire until the subject answered correctly. Upon correct completion of the pre-scan questionnaire the subject faced one block of the task, using a trackball similar to the one used inside the scanner, in order to get used to the experiment’s interface. This block was identical across subjects. We added an English translation of both the experimental instructions and the pre-scan questionnaire to the Supplementary Materials.

Additional evidence that the subjects understood the task can be taken from the data itself. First, as recommended by Bronars (1987) we compared subjects’ choices to simulated decision makers who choose randomly (uniformly) over the budget line. Our subjects exhibited substantially fewer GARP violations and much lower inconsistency indices than would be expected in case of random behavior. We present this result in Fig. 3a. Second, a subject who violates first-order stochastic dominance (FOSD) allocates more tokens to the expensive account, rather than the cheaper account. In our dataset, we found a total of 337 trials (9.5% of trials) with FOSD violations. However, if we allow the subject to allocate the cheaper product up to 90% of her allocation to the expensive product, then the number of FOSD violations drops significantly to 66 trials (1.8% of trials). This result was added to the Supplementary Materials.

Third, as shown below (the horizontal axis stands for the logarithm of the price ratio while the vertical axis stands for the share of the endowment allocated for account Y) all our subjects followed the law of demand, and were sensitive to changes in price ratios (the budget lines slopes), as the share of tokens allocated to the cheaper product increases as its price declines. The individual scatterplots are shown in Supplementary Fig. 7.

Finally, and maybe the most convincing argument, behavior within the fMRI is not substantially different from behavior in the standard behavioral laboratory setting. A comparison with the Choi et al. (2007) study reveals that the distributions of the Afriat inconsistency index are quite similar (Fig 3d in the paper).

Hence, we believe that all of the above indicates that subjects understood the task, even though it was more challenging than most previous MRI tasks.

As to the specific question about corner (extreme) choices - across subjects, 656 choices out of a total of 3,529 trials (18.6% of trials) were “strict” corner choices where subjects allocated all their endowment to one of the accounts. If we loosen our definition for a corner solution, and consider all choices where the subject allocated a maximum of 1 token to one account, and the rest of her endowment to the other account, then the number of corner choices across subjects rises to 759 trials (21.5% of trials). This demonstrates that in most cases, subjects did not choose in the corners and conducted some sort of a tradeoff between the accounts. Note that corner choices are natural for subjects with willingness to accommodate risky choices, especially if the price of one account is significantly more attractive than the price of the other account. Qualitatively, one can see from the graphs above describing the individual level choices that most choices were not “corner” choices (choices on the vertical lines through zero or one).

- There are several missing citations, which I think would require changing some of the language regarding the paper being the “first” in a line of search. Most prominently, it was strange to not see a discussion of Padoa-Schioppa (2013) in Neuron. That paper is not human data, but it is OFC data and focused exclusively on choice variability in the same choice sets. Similarly, Polania et al. (2015) in Nature Communications. This is human data and very much is focused on understanding the neural

mechanisms of choice accuracy/variability. A recent theoretical review in *Nature Neuroscience* (Hayden and Hunt) also gets directly at underlying mechanisms for value-based choice and hence choice inconsistency. Another paper involving fMRI and human data worth citing and possibly discussing is Grueschow et al (2015) in *Neuron*. In particular, that paper discusses posterior cingulate cortex, which is absent in the current paper.

Response:

We agree, and do apologize for that. We have incorporated these insightful works into the manuscript. We also softened the writing and omitted the sentence stating that we are first.

Padoa-Schioppa (2013) shows that firing rate variability of OFC neurons of monkeys before an offer was presented, predicted choice of near-indifferent alternatives. We now cite these findings as evidence that neural variability is related to choice variability, which is consistent with our finding that neural variability accounts for choice inconsistency. Moreover, this is consistent with our finding of a limited role for noise in motor output or imprecision in the numerical representation in generating inconsistent choices. The deviations from consistency in our experiment were observed in valuation regions, suggesting that the value of choice options might be fluctuating on a trial-by-trial basis. These points are now added to the Discussion section.

We also added to this section a short discussion regarding Polania et al. (2015). We suggest that their findings are compatible with our proposed mechanism for choice behavior. We suggest choices are highly inconsistent (as Polania et al. (2015) report) because of the limited ability of the brain to compute the valuations necessary for consistent choice when they are disrupted experimentally. When these regions are intact (as in our study), value signals can be computed, but with a degree of variability inherent to neural computation.

Hunt & Hayden (2017) and Grueschow et al. (2015) are cited as examples for papers discussing the neural mechanism of value computation, and the different regions which take part in the process. Specifically, Hunt & Hayden (2017) was also incorporated into the results section of the new PPI analysis we have added to the manuscript. Moreover, we have expanded the scope of the Discussion and discussed the role of attention in choice variability, and therefore included Denfield et al. (2018), Briggs et al. (2013) and Herrero et al. (2013) in the manuscript as well.

Regarding the PCC: This is a very important comment made by the Reviewer while referencing to Grueschow et al. (2015), one we have neglected in the original manuscript. The PCC is indeed regarded as value-encoding region (see Clithero & Rangel, 2014, for a review), and should have been incorporated in our analyses. We therefore, conducted an additional ROI analyses with a PCC masking. We found similar results to the vmPFC and dACC as the activation in the PCC was significantly correlated with Trial-specific-MMI ($p < 0.0005$, cluster-size correction) and SV ($q(\text{FDR}) < 0.05$). Note, however, that the PCC did not show up in the whole brain analysis, probably due to lack of power. To make sure that not every ROI analysis would give positive results, we also ran the same RFX-GLM using a V1 mask, and got null results. This might suggest that only value-related regions are also involved with choice inconsistency. See the figure below for the neural correlates of Trial-specific-MMI in the PCC and the null result in V1. We added these analyses and results to the ROI analysis section in the manuscript, as well as the subject-level analysis.

4. While the NRUM model presents an account for how the choice inconsistency can arise, I was still left wondering if what is being shown is choice difficulty (e.g. as Shenhav et al. in 2014 showed) in ACC. Using the fitted utility functions, it should be possible to set up an alternative GLM with this as a parametric regressor. Have the authors tried this? It would strengthen their claim(s) to be able to really rule out choice difficulty.

and

As I mentioned earlier, I like the task. It differs from most neuroeconomics fMRI tasks. However, what exactly does it mean for the choice to be “difficult”? It is easy in a standard 2AFC task. It is not so obvious to me that it is the slope of the line, or rather that the slope can be interpreted as a continuous measure of choice difficulty. How do the authors think of choice difficulty here?

Response:

We thank the reviewer for pointing out the role of choice difficulty and for suggesting a solution. Indeed, we can index difficulty in this continuous task and we find our results are robust when controlling for this index. We refer the reader to our response to Reviewer 1 (Major comments, 1b) who raised a similar concern.

5. **Figure 5ABC is helpful. I wonder if it should come earlier in the paper, so that the reader understands the implications, before seeing the fMRI results? I found Figure 5D a bit more confusing. I'd suggest a separate figure for $N>2$.**

Response:

We thank the reviewer for the suggestion. We have made some clarifications regarding the model in the “A model of valuation and inconsistent choices” section and we have adjusted the Introduction so that the reader has some context for the predictions of the “variability in valuation” hypothesis before seeing the fMRI data. However, we believe that moving the model and figure earlier in the paper might disrupt the flow of the paper, and take the figure out of the context of the empirical results. We hope that our adjustments to the introduction in this revision, including more emphasis on the contribution to economics (as requested by Reviewer 1) helps put the fMRI results in context. If the reviewer feels strongly about this request, we will make the necessary adjustments. Regarding the request to separate the figure into two figures - the $n=2$ case and the $n>2$ case: We agree with the reviewer and followed his/her advice (now Figures 5 & 6).

6. **The authors used BrainVoyager for fMRI analysis. The MMI was listed as the first regressor. Does the order matter here? What if SV had been first? Basically, I am asking about orthogonality and/or any other partitioning of the variance that might be assumed in the model. Please list these assumptions, if there are any.**

Response:

The BrainVoyager software runs the GLM using the usual linear model assumptions, and computes the Least Squares predictors for each subject in the first-level analysis. For the second-level group analysis we used the random-effects model (RFX), which uses the first-level estimated beta predictors as the new dependent variable (instead of the BOLD signal), and test their mean value against zero¹. Therefore, the software does not use a “stepwise regression” method, and hence, the order of predictors does not matter here. Had the SV predictor been first, the results would have been exactly the same.

Regarding the issue of orthogonality, in order not to repeat ourselves again, we refer the reviewer to our elaborate response to Reviewer 1 Minor comment 7 where we report on the results of an orthogonality test between trial-specific-MMI and SV.

Minor Comments:

1. **I appreciated the detailed plots presented in the supplement. However, some sort of figure or table summarizing the utility parameters recovered should be placed in the main paper. I think Choi et al have some scatter plots and tables. This is important, as the reader should be able to quickly know if most subjects were risk-averse, risk-neutral, etc.**

¹ See <https://support.brainvoyager.com/brainvoyager/functional-analysis-statistics/43-multi-study-glm/248-users-guide-random-effects-rfx-group-analysis> to elaborate on this topic.

Response:

We completely agree and we are happy to comply. We have added panel c to Figure 3 in the main text (see below), which is similar to figure 6 in Choi et al. (2007). The x-axis describes β , the Disappointment Aversion parameter, while the y-axis describes ρ , the Relative Risk Aversion parameter. Notice that when $\beta < 0$, the subject over-weights the higher prize ("elation seeking", 4 subjects). When $\beta > 0$, the subject over-weights the lower prize 0 ("disappointment aversion", 29 subjects). Higher ρ values represent higher risk aversion levels. When $\beta = 0$, the utility function becomes the common Expected Utility (if, in addition, $\rho = 0$, it is the special case of Expected Value or risk neutrality). We also added Supplementary Table 6, which details the recovered parameters of each and every subject in our sample (see Appendix D to this letter).

- The Figure 1 caption (lines 3,4 in the caption) is written in a confusing way. The parentheses around MMI_D and MMI_{D-i} could be interpreted as a function. I read it this way the first time. I suggest rewriting the sentence to avoid needing the parentheses.

Response:

Thank you for pointing this issue to us. This is now written: "For each observation i the index is the difference between the aggregate index MMI_D calculated for the entire dataset D and the aggregate index MMI_{D-i} calculated for the partial dataset D_{-i} . Formally, the index for observation i is $MMI_D - MMI_{D-i}$ ".

In addition, for clarity, we switched panels A and B and dropped panel C that is relevant only towards the end of the paper and appears in Halevy et al (2018). We hope that now the caption is clearer.

Concerns raised by Reviewer 3:Major comments

- The core result of this study is that both subjective value and the severity of choice inconsistency are correlated with activations in VMPFC and ACC, as identified by parametric regression and conjunction

analyses. This conclusion would be justified if value and inconsistency estimates predicted VMPFC/ACC activity independently. However, this is likely not the case. I suspect a certain degree of collinearity between the value and trial-specific MMI regressors because violations of GARP in trials with high-value alternatives (or large differences in value between both alternatives, respectively) will contribute more to the aggregate extent of inconsistency than GARP violations in low-value alternatives. Since the trial-specific MMI index is the difference between the aggregate MMI and the MMI in the partial dataset when applying a leave-one-out procedure, inconsistent choice should be associated with larger trial-specific MMIs in high value trials than in low value trials. In other words, value estimates and the severity of choice inconsistency indices should correlate. This raises the question if the authors' conjunction analysis really revealed a true overlap of two independently identified activations, or whether the authors merely identified the shared variance between both regressors. The latter possibility would represent a strong challenge to their main conclusion. The authors need to specify to what extent their value estimates and the trial-specific MMI estimates are correlated. If they are correlated, they need to run additional analyses to address this collinearity issue. They could either reduce the collinearity between both regressors by orthogonalizing them (but they need to be careful when interpreting their results as orthogonalization usually assigns the shared variance to one regressor, and the residual variance to the other), or by identifying the unique variance associated with one regressor after partializing out the variance explained by the second regressor, and repeating this procedure with the second regressor after partializing out the variance explained by the first regressor.

Response:

We thank the reviewer for asking us to clarify the results we describe in this paper. In order not to repeat ourselves we refer the reviewer to our response to Minor comment 7 of Reviewer 1 in which we thoroughly address the issue of orthogonality and conduct additional analyses which dissociate between SV and trial-specific MMI.

2. **The authors propose that inconsistent choice behavior is due to neuronal noise that induces variability in the computation of value. Even though I am generally sympathetic to this idea, the authors need to be careful to avoid overly strong claims not supported by their data, and they should consider other interpretations, too. The problem is that the authors measured variability in choice/BOLD, but not true neural noise per se. This means that they cannot rule out that trial-by-trial variability in value might actually not be due to neuronal noise, but arises from other processes, such as the use of cognitive heuristics, fluctuations in attention etc. that might also explain variability in value, and, by extension, the random term parameter e in their random utility model on p. 11. Some people go as far as stating that there is no such thing as neuronal noise. The authors need to state that, although their data are consistent with their noise model, they might be consistent with other accounts, too. Hence, they do not provide direct evidence for their model, but their model merely offers one of many explanations of the variability in value. I need to add that toning down their claims does not limit the originality of their data, it just calls for necessary caution in data interpretation.**

Response:

We thank the reviewer for this important point. We now state explicitly in the Discussion that the sources of variability captured by the error term in the model can arise from many sources including higher order cognitive processes such as fluctuations in attention or heuristics or lower-level process like neuronal noise.

In addition, we have made an effort throughout the manuscript to treat the source of variability in valuations agnostically (this is why we now refer to it as "variability" rather than "neural noise"). Our claim is that variability in value computation areas of the brain induces variability in valuation and choice and hence induces inconsistencies. In our opinion, the possible sources of this noise are almost endless (Faisal, Selten & Wolpert, 2008 and Glimcher 2005) - From quantum variations of electrons, through noise in the interactions of proteins, the probabilistic nature of ion channels, the exact amount of neurotransmitter secreted into the synaptic cleft, etc. However, we agree that it might be the case that such variability also originates from more higher order cognitive processes, such as changes in attention (Denfield et al., 2018; Briggs et al., 2013; Herrero et al., 2013) or employing a given variable heuristic.

As suggested by the reviewer, there are several studies demonstrating an intimate relationship between alterations in attentional levels and neural variability, which we address and cite a few of them in our paper (Denfield et al., (2018); Briggs et al., (2013); Herrero et al. (2013)). However, if a subject employed a heuristic

for making her choices in our task, then it should not have correlated with the trial-specific-MMI, SV, and brain activity. Presumably, the “level” of heuristic should be constant in all trials irrespective if that trial was actually consistent or not, or irrespective of the actual SV of the chosen bundle. Of course, we can’t rule out the possibility that heuristics fluctuate trial-by-trial, but that would also limit the ability of heuristics as an explanatory tool.

In the updated manuscript, we have also ruled out some additional alternative explanations. First, we control for motor and numerical imprecision by using two functional localizers that were collected at the end of the main experiment. Second, we control for choice difficulty by developing a novel index that measures the average distance (in utility) between each bundle and the best bundle (see a detailed description in the paper and in our response to major comment 1b of reviewer 1). Third, we control for confidence in decision making by following Leberton et al. (2015) in using the second order polynomial of the subjective value as a proxy for confidence. As we report in the section “Ruling out alternative explanations”, none of these controls changes our main findings. While these analyses do not exhaust all possible cognitive processes that may affect choice consistency, we believe that ruling out those phenomena strengthens the plausibility of our proposed explanation.

Also, our interpretation is consistent with some important works. In the Discussion section towards the end of our manuscript we quote Fox et al. (2007) stating that: “inconsistency in perception or performance should not be automatically attributed to fluctuations in task-related cognitive processes such as attention but could also be due to ongoing fluctuations in intrinsic neuronal activity” (p.180). In addition, we claim that our results are consistent with Padoa-Schioppa (2013) who demonstrated that neuronal variability is related to choice variability. Finally, we describe the results reported in Drugowitsch et al. (2016) - 89% of the deviations from optimal choice (in their task) can be attributed to errors in value inference, rather than sensory processing or action selection.

Minor Comments:

1. The authors’ value/MMI-related VMPFC cluster identified in their whole-brain GLM is quite dorsal and, in fact, more within medial than ventromedial PFC. A quick Neurosynth search revealed that this region is not typically associated with value, but with a number of other processes instead (default mode network, memory, executive functions etc.). How do the authors reconcile their medial PFC result with their value claims?

Response:

In the whole-brain analysis, our results are indeed more dorsal than usually reported, and hence we changed the text to state mPFC accordingly. However, importantly, in our ROI analysis, we find a significant activation in the dorsal part of the vmPFC ROI that we defined based on the ROI reported in Bartra et al. (2013), which used more than 200 fMRI papers to define the value related activation in the vmPFC. Therefore, we think that in the ROI analyses it is suitable to refer to the activated region as part of the vmPFC.

2. Discussion, p. 16: the abbreviation DDM should be defined on first use.

Response:

Thank you. Done.

Appendix A – individual subjects’ plots – RT against quantiles of the choice difficulty index

These plots present, for each subject, the mean RT (sec) by the choice difficulty index quantiles. Error bars indicate standard deviation.

Appendix B – individual subjects plots – trial-specific-MMI against SV

Plots present for each subject, scatters of Trial-specific-MMI and SV with a linear fit.

Appendix C – Individual distributions of correlation between the simulated data and actual choice data

Subject-specific correlation distributions of simulated noise element \tilde{e} of NRUM with actual Trial-specific-Afriat.

Appendix D – individual recovered utility parameters

Subject	β	ρ
103	0.924	0.241
104	0.020	0.312
202	0.287	0.237
203	0.664	0.491
204	0.096	0.135
205	-0.264	0.435
206	0.476	0.304
401	0.086	0.308
402	-0.211	0.529
403	2.520	0.011
404	0.131	0.025
405	0.111	0.616
406	0.287	0.345
407	0.624	0.232
408	0.105	0.388
409	1.225	0.213
410	0.067	0.750
412	0.209	0.267
413	0.321	0.906
414	1.428	0.059
415	0.269	0.377
416	0.408	0.359
417	0.770	0.024
418	0.667	0.194
419	-0.051	0.432
420	0.257	0.283
421	-0.389	0.813
422	1.888	0.069
424	0.085	0.626
426	0.194	0.357
427	0.370	0.739
428	0.555	0.015
430	0.940	0.030

REVIEWERS' COMMENTS:

Reviewer #1 (Remarks to the Author):

Overall, this response is very thorough and thoughtful, and the authors have addressed my initial concerns. It is unfortunate that the results do not hold when accounting for RT, but the authors' justification makes sense in this situation.

Reviewer #2 (Remarks to the Author):

The authors have done an excellent job addressing all three sets of reviewer comments.

I have two small notes.

1) I believe there is some confusion regarding the usage of "utility" and "random utility" when discussing Fig 5 and 6 (including the captions). Please go back through all usages of those phrases and make sure they are clearly and consistently used.

2) I would like to know what the relationship between RT and "confidence" (as defined following the Lebreton paper). The authors could add this to the new paragraph that starts on line 483.

Reviewer #3 (Remarks to the Author):

The authors have addressed all my comments, I am satisfied with their revisions. This is a nice and interesting paper that will be of interest to a broad readership.